# Rough Transformers: Lightweight and Continuous Time Series Modelling through Signature Patching

**Fernando Moreno-Pino**[1,*]  **Álvaro Arroyo**[1,2,*]  **Harrison Waldon**[1,*]
**Xiaowen Dong**[1,2]  **Álvaro Cartea**[1,3]

[1] Oxford-Man Institute, University of Oxford
[2] Machine Learning Research Group, University of Oxford
[3] Mathematical Institute, University of Oxford

## Abstract

Time-series data in real-world settings typically exhibit long-range dependencies and are observed at non-uniform intervals. In these settings, traditional sequence-based recurrent models struggle. To overcome this, researchers often replace recurrent architectures with Neural ODE-based models to account for irregularly sampled data and use Transformer-based architectures to account for long-range dependencies. Despite the success of these two approaches, both incur very high computational costs for input sequences of even moderate length. To address this challenge, we introduce the Rough Transformer, a variation of the Transformer model that operates on continuous-time representations of input sequences and incurs significantly lower computational costs. In particular, we propose *multi-view signature attention*, which uses path signatures to augment vanilla attention and to capture both local and global (multi-scale) dependencies in the input data, while remaining robust to changes in the sequence length and sampling frequency and yielding improved spatial processing. We find that, on a variety of time-series-related tasks, Rough Transformers consistently outperform their vanilla attention counterparts while obtaining the representational benefits of Neural ODE-based models, all at a fraction of the computational time and memory resources.

## 1 Introduction

Real-world sequential data in areas such as healthcare [65], finance [36], and biology [28] often are irregularly sampled, of variable length, and exhibit long-range dependencies. Furthermore, these data, which may be drawn from financial limit order books [8] or EEG readings [85], are often sampled at high frequency, yielding long sequences of data. Hence, many popular machine learning models struggle to model real-world sequential data, due to input dimension inflexibility, memory constraints, and computational bottlenecks. Rather than treating these data as *discrete* sequences, effective theoretical models often assume data are generated from some underlying *continuous-time* process [53, 66]. Hence, there is an increased interest in developing machine learning methods that use *continuous-time* representations to analyze sequential data.

One recent approach to modelling continuous-time data involves the development of continuous-time analogues of standard deep learning models, such as Neural ODEs [12] and Neural CDEs [45], which extend ResNets [37] and RNNs [30], respectively, to continuous-time settings. Instead of processing discrete data directly, these models operate on a latent continuous-time representation

---

*Equal contribution.
  Email: {fernando.moreno-pino, alvaro.arroyo}@eng.ox.ac.uk
  Code available at: https://github.com/AlvaroArroyo/RFormer

38th Conference on Neural Information Processing Systems (NeurIPS 2024).

of input sequences. This approach is successful in continuous-time modelling tasks where standard deep recurrent models fail. In particular, extensions of vanilla Neural ODEs to the time-series setting [70, 45] succeed in various domains such as adaptive uncertainty quantification [59], counterfactual inference [79], or generative modelling [7].

In many practical settings, such as financial market volatility [20, 54] or heart rate fluctuations [35], continuous-time data also exhibit long-range dependencies. That is, data from the distant past may impact the system's current behavior. Deep recurrent models struggle in this setting due to vanishing gradients, whereas continuous-time analogues of these models have been shown to address this difficulty [46]. Several recent works [52, 58] also successfully extract long-range dependencies from sequential data with Transformers [86], which learn temporal dependencies of a tokenized representation of input sequences. Extracting such temporal dependencies requires a positional encoding of input data, because the attention mechanism is permutation invariant, which projects data into some latent space. The parallelizable nature of the Transformer allows for rapid training and evaluation on sequences of moderate length and it contributes to its success in fields such as natural language processing (NLP).

While the above approaches succeed in certain settings, several limitations hinder their wider applications. On the one hand, Neural ODEs and their analogues [45, 70] bear substantial computational costs when modelling long sequences of high dimension; see [57]. On the other hand, Transformers operate on discrete-time representations of input sequences, whose relative ordering is represented by the positional encoding. This representation may inhibit their expressivity in continuous-time data modelling tasks [91]. Moreover, Transformer-based models suffer from a number of difficulties, including (i) input sequences must be sampled at the same times, (ii) the sequence length must be fixed, and (iii) the computational cost scales quadratically in the length of the input sequence. These difficulties severely limit the application of Transformers to continuous-time data modelling.

**Contributions 1)** We introduce *Rough Transformers*, a variant of the Transformer architecture amenable to the processing of continuous-time signals, which can be easily integrated into existing code-bases. The Rough Transformer is built upon the path signature from Rough Path Theory [51]. We define a novel, multi-scale transformation which projects discrete input data to a continuous-time path and compresses the input data with minimal information loss. Moreover, this transformation is an efficient feature representation of continuous-time paths, because linear functionals of path signatures approximate continuous functions of paths arbitrarily well (see Theorem A.2 in Appendix A).

**2)** We introduce the *multi-view attention mechanism* to extract both local and global dependencies of very long time-series efficiently. This mechanism operates directly on continuous-time representations of data without the need for expensive numerical solvers or constraints on the smoothness of the data stream. Moreover, the multi-view attention mechanism is provably robust to irregularly sampled data.

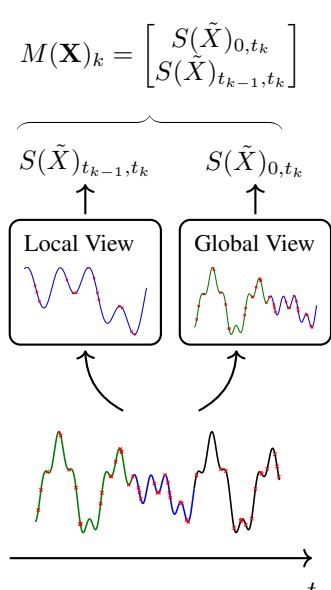

$$M(\mathbf{X})_k = \begin{bmatrix} S(\tilde{X})_{0,t_k} \\ S(\tilde{X})_{t_{k-1},t_k} \end{bmatrix}$$

$S(\tilde{X})_{t_{k-1},t_k}$     $S(\tilde{X})_{0,t_k}$

Local View     Global View

$t$

Figure 1: A representation of the multi-view signature. The continuous-time path is irregularly sampled at points marked with a red $x$. The local and global signatures of a linear interpolation of these points are computed and concatenated to form the multi-view signature. The multi-view signature transform consists of $\overline{L}$ multi-view signatures.

**3)** We carry out extensive experimentation on long and irregularly sampled time-series data. In particular, we show that Rough Transformers (i) improve the learning dynamics of the Transformer, making it more sample-efficient and allowing it to achieve better out-of-sample results, (ii) reduce the training cost by a factor of up to $25\times$ when compared with vanilla Transformers and more when compared with Neural ODE based architectures, (iii) maintain similar performance when data are irregularly sampled, where traditional recurrent-based models suffer a substantial decrease in

performance [70], and (iv) yield improved spatial processing, accounting for relationships between different temporal channels without having to pre-define a specific inter-channel relation structure.

## 2 Background and Methodology

**Problem Formulation.** In many real-world scenarios, sequential data are time-series sampled from some underlying continuous-time process, so datasets consist of long, irregularly sampled sequences of varied lengths. In these settings, the problem of sequence modelling is described as follows. Let $C(\mathbb{R}^+; \mathbb{R}^d) = \{g : \mathbb{R}^+ \to \mathbb{R}^d \mid g \text{ continuous}\}$, and consider $\widehat{X} \in C(\mathbb{R}^+; \mathbb{R}^d)$ which we call a continuous-time *path*. A time-series of length $L$ with sampling times $\mathcal{T}_{\mathbf{X}} = \{t_i\}_{i=1}^{L} \subset \mathbb{R}^+$ is defined as $\mathbf{X} = ((t_1, X_1), ..., (t_L, X_L))$, where $X_i = \widehat{X}(t_i) \in \mathbb{R}^d$. Now, define a continuous function on paths $f : C(\mathbb{R}^+; \mathbb{R}^d) \to \mathbb{R}^k$. Next define a dataset $\mathcal{D} = \left\{ (\mathbf{X}^i, f(\widehat{X}^i))_{i=1}^{N} \right\}$. We seek to approximate the function $f$ from the set $\mathcal{D}$ for some downstream task. Importantly, we do not assume that $\mathcal{T}_{\mathbf{X}} = \mathcal{T}_{\mathbf{Y}}$ for all $\mathbf{X}, \mathbf{Y} \in \mathcal{D}$, so that $\mathcal{D}$ may be irregularly sampled.

**Sequence Modelling with Transformers.** Transformers are used extensively as a baseline architecture to approximate functions of discrete-time sequential data and are successfully applied to settings when input sequences are fixed in length, relatively short, and sampled at regular intervals. First, the Transformer projects input time series $\mathbf{X} \in \mathbb{R}^{L \times d}$ to a high-dimensional space $\mathbf{X} \mapsto T(\mathbf{X}) \in \mathbb{R}^{L \times d'}$ for $d' >> d$ using some linear positional encoding $T : \mathbb{R}^{L \times d} \to \mathbb{R}^{L \times d'}$. Next, a latent representation of the encoded sequence is learned by a multi-headed self-attention mechanism which splits $T(\mathbf{X})$ into $H$ distinct query, key, and value sequences: $Q_h = T(\mathbf{X})W_h^Q$, $K_h = T(\mathbf{X})W_h^K$, $V_h = T(\mathbf{X})W_h^V$, respectively, with $h = 1, ..., H$ and weight matrices $W_h^Q, W_h^K, W_h^V \in \mathbb{R}^{d' \times d'}$. The multi-head self-attention calculation for each head is given by

$$O_h = \text{softmax}\left(\frac{Q_h K_h^\intercal}{\sqrt{d_k}}\right) V_h \,, \tag{1}$$

and the latent representation is projected to the output space $\mathbb{R}^k$ using a multi-layer perceptron (MLP).

The input length $L$ of the MLP and the Transformer is fixed by assumption. To evaluate the Transformer on a time-series $\mathbf{X}$ with $|\mathcal{T}_{\mathbf{X}}| \neq n$, one must perform some transformation (interpolation, extrapolation, etc.) which may degrade the performance of the model. Furthermore, the memory and time complexity of the Transformer is of order $O(L^2 d)$, which presents a substantial difficulty in modelling long sequences.

**Rough Path Signatures.** Broadly, the difficulties faced by the Transformer in modelling time-series stem from time-series being sampled from underlying *continuous-time* objects, while the attention mechanism underpinning the Transformer is designed to model discrete sequences. To address these difficulties, Rough Transformers augment standard Transformers by lifting the input time-series to the space of continuous-time functions and performing the self-attention calculation in this infinite-dimensional space. To achieve this, we use the path signature from Rough Path Theory.

For a continuous-time path $\widehat{X} \in C_b^1(\mathbb{R}^+; \mathbb{R}^d)$ and times $s, t \in \mathbb{R}^+$, the path signature of $\widehat{X}$ from $s$ to $t$, denoted $S(\widehat{X})_{s,t}$, is defined as follows. First, let

$$\mathcal{I}_d = \{(i_1, ..., i_p) : i_j \in \{1, ..., d\} \,\forall j \text{ and } p \in \mathbb{N}\} \tag{2}$$

denote the set of all $d$-multi-indices and $\mathcal{I}_d^n = \{I \in \mathcal{I}_d : |I| = n\}$. Next, set $S(\widehat{X})_{s,t}^0 := 1$ and for any $I \in \mathcal{I}_d$, define

$$S(\widehat{X})_{s,t}^I = \int_{s < u_1 < ... < u_p < t} \dot{\widehat{X}}^{i_1}(u_1) \cdots \dot{\widehat{X}}^{i_p}(u_p) \, du_1 \ldots du_p \,, \tag{3}$$

where $\dot{\widehat{X}}^j = d\widehat{X}^j/dt$. Abusing notation, define level $n$ of the signature as

$$S^n(\widehat{X})_{s,t} = \left\{ S(\widehat{X})_{s,t}^I : I \in \mathcal{I}_d^n \right\} \,. \tag{4}$$

and define the signature as the infinite sequence

$$S(\widehat{X})^n_{s,t} = \left(S(\widehat{X})^0_{s,t}, S(\widehat{X})^1_{s,t}, ..., S(\widehat{X})^n_{s,t}, ...\right). \tag{5}$$

Finally, define the truncation of the signature $S(\widehat{X})^{\leq n}_{s,t} = (S(\widehat{X})^0_{s,t}, ..., S(\widehat{X})^n_{s,t})$, where $S(\widehat{X})^n_{s,t}$ can be interpreted as an element of the *extended tensor algebra* of $\mathbb{R}^d$:

$$T((\mathbb{R}^d)) = \left\{(a_0, ..., a_n, ...) : a_n \in \mathbb{R}^{d \otimes n}\right\}. \tag{6}$$

Analogously, we say that $S(\widehat{X})^{\leq n}_{s,t} \in T((\mathbb{R}^d))_{\leq n}$. A central property of the signature is that is invariant with respect to time-reparameterization [51]. That is, let $\gamma : [0, T] \to [0, T]$ be surjective, continuous, and non-decreasing. Then we have

$$S(\widehat{X})_{0,T} = S(\widehat{X} \circ \gamma)_{0,T}, \tag{7}$$

which will be crucial to demonstrate the Rough Transformer's robustness to irregularly sampled data.

In contrast to wavelets or Fourier transforms, which parameterize paths on a functional basis, the signature provides a basis for functions of continuous paths. Hence, the path signature is well-suited to sequence modelling tasks in which one seeks to learn a function of the underlying functional. For a more rigorous presentation of signatures and a description of additional properties, see Appendix A and Lyons et al. [51].

## 3 Rough Transformers

Now, we construct the Rough Transformer, a Transformer-based architecture that operates on continuous-time sequential data by means of the path signature.

Let $\mathcal{D}$ be a dataset of irregularly sampled time-series. To project a discretized time-series $\mathbf{X} \in \mathcal{D}$ to a continuous-time object, let $\tilde{X}$ denote the piecewise-linear interpolation of $\mathbf{X}$.[2] Next, for $t_k \in \mathcal{T}$, define the *multi-view signature*

$$M(\mathbf{X})_k := \left(S(\tilde{X})_{0,t_k}, S(\tilde{X})_{t_{k-1},t_k}\right). \tag{8}$$

In what follows, we refer to the components $\left(S(\tilde{X})_{0,t_k}, S(\tilde{X})_{t_{k-1},t_k}\right)$ as *global* and *local*, respectively; see Figure 1. Intuitively, one can interpret the global component as an efficient representation of long-term information (see Theorem A.2 in Appendix A), and the local component as a type of convolutional filter that is invariant to the sampling rate of the signal. Now, define the *multi-view signature transform* $M(\mathbf{X}) = (M(\mathbf{X})_1, ..., M(\mathbf{X})_{\bar{L}})$, and denote by $M(\mathbf{X})^{\leq n}$ the truncated signature for a truncation level $n$. Next, define the *multi-view attention mechanism*, which uses the multi-view signature transform to extend the standard attention mechanism to the space of continuous functions [51]. First, fix a truncation level $n \in \mathbb{N}$, and let $\bar{d} \in \mathbb{N}$ be such that $M(\mathbf{X})^{\leq n}_{\bar{k}} \in \mathbb{R}^{\bar{d}}$. For $h = 1, ..., H$ let $W^{\tilde{Q}, \tilde{K}, \tilde{V}}_h \in \mathbb{R}^{\bar{d} \times \bar{d}'}$ for some $\bar{d}' \in \mathbb{N}$, and let

$$\tilde{Q}_h = M(\mathbf{X})^{\leq n} W^{\tilde{Q}}_h, \quad \tilde{K}_h = M(\mathbf{X})^{\leq n} W^{\tilde{K}}_h, \quad \tilde{V}_h = M(\mathbf{X})^{\leq n} W^{\tilde{V}}_h. \tag{9}$$

Then, the attention calculation is given by

$$O_h = \text{softmax}\left(\frac{\tilde{Q}_h \tilde{K}^\intercal_h}{\sqrt{\bar{d}'}}\right) \tilde{V}_h. \tag{10}$$

Notice that the attention calculation is similar to (1), however, we stress that the multi-view attention is built on *continuous-time* objects, the signatures, while the standard attention mechanism acts on discrete objects. The multi-view signature provides a compressed representation of the time series, minimizing the computational costs associated to quadratic scaling without excessive loss of representational capacity, see Appendix F.

---

[2]Any continuous-time interpolation of $\mathbf{X}$ can be used, e.g., splines. However, the signature computation of piecewise-linear paths is particularly fast; see Appendix A.

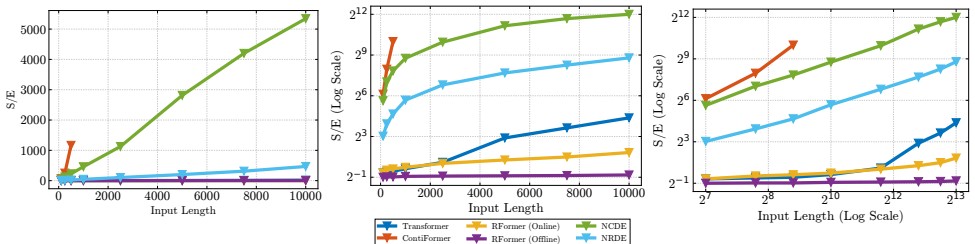

Figure 2: Seconds per epoch for growing input length and for different model types on the sinusoidal dataset. **Left:** Log Scale. **Middle:** Regular Scale. **Right:** Log-log scale. When a line stops, it indicates an OOM error.

## 3.1 Advantages of Rough Transformers

**Computational Efficiency.** As demonstrated in Section 4, multi-view attention mechanism can substantially reduce the computational cost of vanilla Transformers. In particular, the attention calculation decreases from $O(L^2 d)$ in the vanilla case to $O(\overline{L}^2 d)$, where $\overline{L} << L$ with Rough Transformers. This enables both faster wall-clock training time and the ability to process long input sequences which would otherwise yield out-of-memory errors for the vanilla Transformer, see Figure 2. Moreover, the multi-view attention mechanism does not require backpropagation through the signature calculation, which can be computed *offline*. This is significantly more computationally efficient compared with the complexity of computing signatures batch-wise in every training step. Finally, the signature of piecewise-linear paths can be computed explicitly, see Appendix A, and there are a number of Python packages devoted to optimized signature calculation [43, 68].

**Variable Length and Irregular Sampling.** The multi-view signature transform underpinning Rough Transformers is evaluated by constructing a continuous-time interpolation of input data and computing a series of iterated integrals of this interpolation. The bounds of these integrals are a fixed set of time points, meaning that the sequence length of the multi-view attention mechanism is fixed and independent of the sequence length of input samples. Furthermore, the following proposition shows that the output of the Rough Transformer for two (possibly irregular) samplings of the same path is similar.

**Proposition 3.1.** *Let $\mathbb{T}$ be a Rough Transformer. Suppose $\widehat{X} : [0, T] \to \mathbb{R}^d$ is a continuous-time process, and let $\gamma : [0, T] \to [0, T]$ denote a time-reparameterization. Suppose $\mathbf{X}$ and $\mathbf{X}'$ are samplings of $\widehat{X}$ and $\widehat{X} \circ \gamma$, respectively. Then $\mathbb{T}(\mathbf{X}) \approx \mathbb{T}(\mathbf{X}')$.*

*Proof.* By (7), $S(\widehat{X})_{s,t} = S(\widehat{X} \circ \gamma)_{s,t}$ for all $s, t \in [0, T]$. Hence, one has $M(X^1) \approx M(X^2)$. Finally, $\mathbb{T}(X^1) \approx \mathbb{T}(X^2)$ because the attention mechanism and final MLP are both continuous. $\square$

Hence, the Rough Transformer is robust to irregular sampling. In many tasks, the sampling times convey important information about the time-series. In these settings, one may augment the input time-series with its sampling times, that is, write $X = ((t_0, X_0), ..., (t_L, X_L))$.

**Spatial Processing.** While an interpolation of input data could be sampled to make vanilla Transformers independent of the length of the input sequence, important locality information could be lost, see Appendix F.2. Instead, Rough Transformers summarize spatial interactions between channels by means of the multi-view signature transform. One may notice that in (5), the dimension of the signature grows exponentially in the level of the signature $n$. In particular, when $X_i \in \mathbb{R}^d$, $|S(\tilde{X})^{\leq n}_{0,t}| = \frac{d(d^n - 1)}{d - 1} = O(d^n)$, so the multi-view attention calculation is of order $O(\bar{L}^2 d^n)$. In many practical time-series modelling problems, however, the value of $d$ is not very large. The signature terms also decay factorially in the signature level $n$ (see Proposition A.3 in Appendix A), so in practice, one may take the value of $n$ to be small without sacrificing performance. The majority of computational savings result from the reduction of the sequence length to $\bar{L}$, and in practice, we take $\bar{L} << L$.

When the dimension $d$ is large, there are three possible remedies to maintain computational efficiency. First, instead of computing the signature in $M(X)_k = (S(X)_{0,t_k}, S(\tilde{X}_{t_{k-1}, t_k}))$, one may compute

the *log-signature*, which is a compressed version of the signature [67]. When the dimension is large enough such that the log-signature is computationally infeasible, one may instead compute the *univariate* signatures of features coupled with the time channel. That is, consider $\widehat{X} \in C([0, T]; \mathbb{R}^d)$, with $\widehat{X}(t) = (\widehat{X}_1(t), ..., \widehat{X}_d(t))$. Denote the time-added function $\overline{X}_i(t) := (t, \widehat{X}_i(t))$. Then we define the *univariate multi-view signature*

$$\widehat{M}(\widehat{X})_k = \left( M(\overline{X}_1)_k, ..., M(\overline{X}_d)_k \right) . \tag{11}$$

The attention mechanism in this case is constructed as before. Fixing the maximum signature depth to be some value $n^*$, one sees that the number of features in the univariate multi-view signature is approximately $2^{n^*} d$. In practice we find that $n^* \leq 5$ provides sufficient performance, so the order of the attention calculation is $O(C \bar{L}^2 d)$ for $C \leq 2^{n^*}$. Finally, one may use randomized signatures to reduce dimension by using a Johnson-Lindenstrauss-type projection to a low-dimensional latent space and computing the signature in this space, as in [21, 19].

## 4 Experiments

In this section, we present empirical results for the effectiveness of the Rough Transformer, hereafter denoted `RFormer`, on a variety of time-series-related tasks. Experimental and hyperparameter details regarding the implementation of the method are in Appendices C and D. We consider long multivariate time-series as our main experimental setting because we expect signatures to perform best in this scenario. Additional experimentation on long-range reasoning tasks on image-based datasets is left for future work, as these would likely require additional inductive biases.

To benchmark `RFormer`, we consider both discrete-time and continuous-time models. In particular, we include as main baselines traditional RNN models (`GRU` [15]), ODE-based methods designed for sequential data (`Neural-CDE` [45]), as well as ODE-based methods explicitly designed for long time-series (`Neural-RDE` [57]).[3] Furthermore, we compare against a vanilla `Transformer` [86] which is the `RFormer` backbone. Finally, we present comparisons with a recent continuous-time Transformer model, `ContiFormer` [13], to highlight the computational efficiency gap between `RFormer` and similar continuous-time models. We note that the first two tasks focus on evaluating the performance improvement of `RFormer` over the `Transformer` baseline. For other long-range tasks, we include comparisons to recent state-space models [31, 62, 81]. In the irregular sampling regime, we benchmark against state-of-the-art models tailored to that setting [61, 78]. See Appendix B for additional discussion on related models and more details about our experimental choices.

### 4.1 Time Series Processing

**Frequency Classification.** Our first experiment is based on a set of synthetically generated time series from continuous paths of the form

$$\widehat{X}(t) = g(t) \sin(\omega t + \nu) + \eta(t), \tag{12}$$

where $g(t)$ is a non-linear trend component, $\nu$ and $\eta$ are two noise terms, and $\omega$ is the frequency. Here, the task of the model is to classify the time-series according to its frequency $\omega$. We consider 1000 samples in 100 classes with $\omega$ evenly distributed from 10 to 500. Each time-series is regularly sampled with 2000 times-steps on the interval $[0, 1]$.

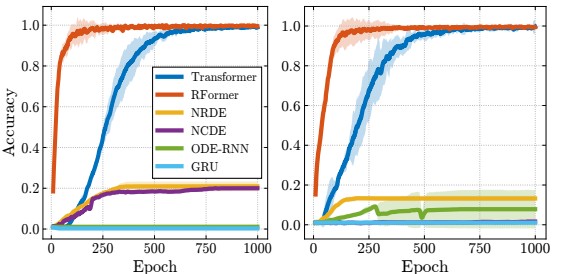

Figure 3: Test accuracy per epoch for the frequency classification task across three random seeds. **Left:** Sinusoidal dataset. **Right:** Long Sinusoidal dataset.

This synthetic experiment is similar to others in recent work on time-series modelling [49, 89, 55]. We include an additional experiment in which we alter the signal in (12) so its frequency is $\omega_0$ for $t < t_0$ and $\omega_1$ afterward, where the task is to classify the sinusoid based on the first frequency. We call this dataset the "long sinusoidal" dataset. This extension of the original experiment aims to test the ability of the model to perform long-range reasoning effectively. Note that for this task, we also add `ODE-RNN` [70] to the previously mentioned baselines.

---

[3]We only benchmark `Neural-CDE` models in settings where time series are of relatively short length, due to the computational demands of this model for longer sequences.

Figure 3 shows that the inclusion of both local and global information with the multi-view signature enhances the sample efficiency of the `RFormer` over the vanilla `Transformer` model, even though the attention mechanism is now operating on a much shorter sequence. When compared with other models, we see that `GRU` and `ODE-RNN` fail to capture the information in the signal, and are not able to obtain any meaningful performance improvement throughout the training period. This highlights the shortcomings of most RNN-based models when processing sequences of moderate length, which are very common in real-world applications. Both `Neural-CDE` and `Neural-RDE` capture some useful dependencies in the time series but fall short compared with both vanilla `Transformer` and `RFormer`.

**HR dataset.** Next, we consider the Heart Rate dataset from the TSR archive [83], originally sourced from Beth Israel Deaconess Medical Center (BIDMC). This dataset consists of time-series sampled from patient ECG readings, and each model is tasked to perform a regression by forecasting the patient's heart rate (HR) at

Table 1: Test RMSE (mean $\pm$ std) computed across five seeds on the Heart Rate (HR) dataset.

| Model | HR RMSE $\downarrow$ |
|---|---|
| ODE-RNN$^\diamond$ | $13.06 \pm 0.00$ |
| Neural-CDE$^\diamond$ | $9.82 \pm 0.34$ |
| Neural-RDE$^\diamond$ | $2.97 \pm 0.45$ |
| GRU$^\dagger$ | $13.06 \pm 0.00$ |
| ODE-RNN$^\dagger$ | $13.06 \pm 0.00$ |
| Neural-RDE$^\dagger$ | $\underline{4.04} \pm 0.11$ |
| Transformer | $8.24 \pm 2.24$ |
| ContiFormer | OOM |
| **RFormer** | $\mathbf{2.66 \pm 0.21}$ |

the sample's conclusion. The data, sampled at 125Hz, consists of three-channel time-series (including time), each spanning 4000 time steps. We used the L2 loss metric to assess the performance. Table 1 shows the results, where $\diamond$ denotes the results from Morrill et al. [57] and $\dagger$ our reproduction. The sequences in the HR dataset are sufficiently short to remain within memory when running the `Transformer` model. The baseline `Transformer` model improves over `GRU`, and `ODE-RNN`, however, it is less competitive when compared with `Neural-RDE`, suggesting that the Transformer is not particularly well-suited for this type of task. However, the `RFormer` model improves over the baseline `Transformer` by 67%. Across all tasks, we see significant improvements in efficiency as a consequence of the signature computation. We elaborate on this in more detail in the following subsection.

**Long Time Series Classification.** We now evaluate the performance of `RFormer` on five long time series classification tasks from the UEA time series classification archive [3]. A summary of these datasets is provided in Table 13 in Appendix E. As previously done in [57], the original train and test datasets are merged and then randomly divided into new train, validation, and test sets, following a 70/15/15 split. The resulting performance metrics are summarized in Table 2.[4]

In this setting, we see that `RFormer` generally matches or slightly outperforms the continuous-time and SSM baselines. Due to the scaling problems of `ContiFormer` with respect to sequence length, we were unable to run this baseline within GPU memory constraints in most cases, and thus no results are reported (see Appendix G.2 for efficiency comparisons between models). In contrast, `RFormer` can cheaply train on the same device (see Section 4.2 for details) due to its ability to take advantage of the parallel nature of GPU processing and compress the original time series. This is especially noticeable when compared to continuous-time models (`Neural-CDE`, `Neural-RDE`, `LogCDE`), which are sometimes orders of magnitude slower than our model and consistently report lower or similar results. Additional experimental details can be found in Appendix G, as well as some experiments on hyperparameter sensitivity.

Table 2: Classification performance on various long context temporal datasets from UCR TS archive.

| Dataset | LRU | S5 | S6 | Mamba | NCDE | NRDE | LogNCDE | Transformer | RFormer |
|---|---|---|---|---|---|---|---|---|---|
| SCP1 | $82.6 \pm 3.4$ | $\mathbf{89.9 \pm 4.6}$ | $82.8 \pm 2.7$ | $80.7 \pm 1.4$ | $79.8 \pm 5.6$ | $80.9 \pm 2.5$ | $83.1 \pm 2.8$ | $84.3 \pm 6.3$ | $81.2 \pm 2.8$ |
| SCP2 | $51.2 \pm 3.6$ | $50.5 \pm 2.6$ | $49.9 \pm 9.5$ | $48.2 \pm 3.9$ | $53.0 \pm 2.8$ | $\mathbf{53.7 \pm 6.9}$ | $53.7 \pm 4.1$ | $\underline{49.1 \pm 2.5}$ | $52.3 \pm 3.7$ |
| MI | $48.4 \pm 5.0$ | $47.7 \pm 5.5$ | $51.3 \pm 4.7$ | $47.7 \pm 4.5$ | $\underline{49.5 \pm 2.8}$ | $47.0 \pm 5.7$ | $53.7 \pm 5.3$ | $50.5 \pm 3.0$ | $\mathbf{55.8 \pm 6.6}$ |
| EW | $87.8 \pm 2.8$ | $81.1 \pm 3.7$ | $85.0 \pm 16.1$ | $70.9 \pm 15.8$ | $75.0 \pm 3.9$ | $83.9 \pm 7.3$ | $\underline{85.6 \pm 5.1}$ | OOM | $\mathbf{90.3 \pm 0.1}$ |
| ETC | $\underline{21.5 \pm 2.1}$ | $24.1 \pm 4.3$ | $26.4 \pm 6.4$ | $27.9 \pm 4.5$ | $29.9 \pm 6.5$ | $25.3 \pm 1.8$ | $34.4 \pm 6.4$ | $\mathbf{40.5 \pm 6.3}$ | $34.7 \pm 4.1$ |
| HB | $\mathbf{78.4 \pm 6.7}$ | $\underline{77.7 \pm 5.5}$ | $76.5 \pm 8.3$ | $76.2 \pm 3.8$ | $73.9 \pm 2.6$ | $72.9 \pm 4.8$ | $75.2 \pm 4.6$ | $70.5 \pm 0.1$ | $\underline{72.5 \pm 0.1}$ |
| Av. | 61.7 | 61.8 | 62.0 | 58.6 | 60.2 | 60.6 | $\underline{64.3}$ | 59.0 | $\mathbf{64.5}$ |

---

[4]We note that baseline results for this task were taken from [87].

## 4.2 Training Efficiency

Here, we focus on the computational gains of the model when compared with vanilla Transformers and methods that require numerical ODE solvers.

Attention-based architectures are highly parallelizable on modern GPUs, as opposed to traditional RNN models which require sequential updating. However, vanilla attention experiences a bottleneck in memory and time complexity as the sequence length $L$ grows. As covered above in Section 3, variations of the signature transform allow the model to operate on a reduced sequence length $\bar{L}$ without increasing the dimensionality in a way that would become problematic for the model. This allows us to bypass the quadratic complexity of the model without resorting to sparsity techniques commonly used in the literature [26, 49].

Tables 1-3 show that `RFormer` is competitive when modelling datasets with extremely long sequences without an explosion in the memory requirements. `RFormer` exploits the parallelism of the attention mechanism to significantly accelerate training time, as the length of the input sequence is decreased substantially. In particular, we observe speedups of $1.4\times$ to $26.11\times$ with respect to standard attention, and higher when compared with all methods requiring numerical solutions to ODEs. The computational efficiency gains of `RFormer` are attained due to the signature transform reducing the length of the time-series with minimal information loss. The effectiveness of this transformation can be seen from the ablation study carried out in Appendix F. This contrasts with NRDEs [57], which augment NCDEs with local signatures of input data, and find that smaller windows often perform better. Furthermore, NRDEs do not experience the same computational gains as `RFormer` because they must perform many costly ODE integration steps.

Table 3: Seconds per epoch for all models considered.

| Model | Sec. / Epoch | | |
|---|---|---|---|
| | **Sine** | **EW** | **HR** |
| GRU | **0.12** | 0.25 | 1.07 |
| ODE-RNN | 5.39 | 48.59 | 50.71 |
| Neural-CDE | 9.83 | - | - |
| Neural-RDE | 0.85 | 5.23 | 9.52 |
| Transformer | 0.77 | OOM | 11.71 |
| **RFormer** | 0.55 | **0.11** | **0.45** |
| **Speedup** | $1.4\times$ | - | $26.11\times$ |

Table 4: Dataset processing times for training, validation, and testing phases.

| Dataset | **Train** | **Val** | **Test** |
|---|---|---|---|
| Eigenworms | 1.11 s. | 0.19 s. | 0.19 s. |
| HR | 4.23 s. | 0.84 s. | 0.85 s. |
| Sine (1k) | 0.39 s. | 0.39 s. | 0.39 s. |
| Sine (5k) | 0.51 s. | 0.51 s. | 0.51 s. |
| Sine (20k) | 1.64 s. | 1.64 s. | 1.64 s. |
| Sine (100k) | 5.74 s. | 5.74 s. | 5.74 s. |

In Figure 2, we showcase the improvements in computational efficiency of `RFormer` compared to vanilla Transformers [86], continuous-time Transformers [13], and other continuous-time RNNs [45, 57] when processing sequences from $L = 100$ samples up to $L = 10K$. As seen, `RFormer` is significantly more efficient than its continuous-time and vanilla counterparts, even when performing the signature computation online, which involves computing the signatures for each batch during training, resulting in significant redundant computation. When signatures are precomputed just once before training, the computational time of each epoch remains *constant* across input all sequence lengths including $L = 10K$ (see the exact signature computation times for different datasets in Table 4). We also stress the fact that `RFormer` also scales gracefully for extremely long sequences (up to $L = 250K$) with both online and offline computation of the signatures, as shown in Appendix G. Finally, we highlight that `ContiFormer` has a sample complexity of $\mathcal{O}(L^2 d^2 S)$, where $S$ represents the normalized number of function evaluations of the numerical ODE solver, which makes `ContiFormer` orders of magnitude more computationally intensive when compared to `RFormer` and prevents the model from running on sequences longer than 500 points without running out of memory (see device details in Appendix C).

## 4.3 Irregular Time Series Classification

So far, we mainly focused on the efficiency and inductive bias afforded to the model through the use of signatures. However, a key element of `RFormer` is that it can naturally deal with irregularly sampled sequences without expensive numerical ODE solvers. This property follows from the fact that signatures are *invariant to time reparameterization*, see Proposition 3.1. In this subsection, we empirically test this property by training the model on the same datasets but randomly dropping a percentage of the data points. This test intends to find if the model is able to learn continuous-time representations of the original input time-series. The results can be found in Table 5. We find that

`RFormer` consistently results in the best performance, with a small performance drop when compared to the full dataset. Importantly, this property is achieved in conjunction with the efficiency gains afforded to the model and without the use of expensive numerical ODE solvers.[5] Finally, we perform an additional set of experiments on the 15 univariate classification datasets from the UEA time series classification archive and compare our model with recent state-of-the-art models for irregular time series [61, 78]. Across the board, we find that our model is both faster and more accurate than the continuous-time benchmark *despite having a discrete-time Transformer backbone*, as shown in Figure 4, which introduces Continuous Recurrent Units (CRU) [78] as an extra baseline. For more details and more exhaustive experimentation on random data drops, see Appendix G.

Table 5: Performance of all models under a random 50% drop in datapoints per epoch.

| Model | 50% Drop Performance | | | |
|---|---|---|---|---|
| | EW (%) ↑ | HR ↓ | Sine (%) ↑ | Sine Long (%) ↑ |
| GRU | 35.90 | 13.06 | 0.96 | 1.16 |
| ODE-RNN | 37.61 | 13.06 | 1.06 | 1.23 |
| Neural-RDE | 60.68 | 4.67 | 0.94 | 0.87 |
| Transformer | OOM | 12.73 | 7.37 | 20.23 |
| **RFormer** | **87.69** | **2.96** | **59.57** | **93.17** |

## 5   Reasons for improved model performance

In this final section, we provide explanations for the superior inductive bias of the `RFormer` model compared to its vanilla Transformer counterpart, despite its lower computational cost.

### 5.1   Spatial Processing

First, we highlight that a key reason the model achieves significant compression benefits in the tasks considered is its ability to *jointly* account for temporal and spatial interactions through the self-attention mechanism and signature terms, respectively. In particular, we believe that for certain datasets, the relationships between different channels of the time series may hold more importance than the temporal information itself, which can often be redundant. This is exemplified in the `Eigenworms` dataset, which experiences a  20% performance drop when employing univariate signatures, but is able to achieve state-of-the-art performance with a $600\times$ compression rate in the temporal dimension when signatures are applied across all channels, as shown in Figure 6. To this end, we draw parallels between the use of signatures and the field of temporal graph processing, where the use of the signature over all channels can be seen as a fully connected graph, capturing information from all channels, and the univariate signature would correspond to a graph with only self-connections between the nodes, as depicted in Figure 5. In our view, this hints towards the idea of using sparse graph learning techniques [17, 23] to reduce the explosion in signature terms while retaining the ability to perform effective spatial processing.

To empirically test these claims, we design a synthetic experiment using a 2-channel time series. Each channel contains a signal of the form $\sin(\omega_i t + \nu_i), i = 1, 2$, where $\omega_i$ and $\nu_i$ are randomly sampled from the interval $[0, 2\pi]$. For half of the dataset, the last 1% of temporal samples in the second channel are set to match the frequency of the first channel. The task is to classify whether the samples in this final interval are of the same frequency. As shown in Figure 5, `RFormer` demonstrates greater sample efficiency and achieves higher test accuracy compared to its vanilla Transformer counterpart, highlighting the effectiveness of signatures in spatial processing.

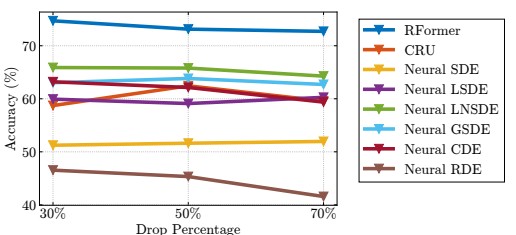

Figure 4: Average performance of all models on the 15 univariate datasets from the UEA Time Series archive under different degrees of data drop.

---

[5]We use our own reproduction to test the performance of all models in irregularly sampled datasets. Random dropping requires window sizes larger than 2 because signatures cannot be computed over a single point. The best step size was chosen in accordance with performance on the validation dataset, see Appendix D.

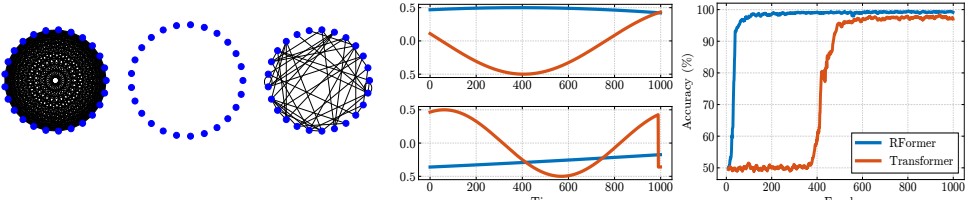

Figure 5: **Left:** Graph connectivity structures for multivariate, univariate and sparse signature. **Middle:** Example samples for synthetic task. **Right:** Performance on spatial synthetic experiment.

## 5.2 Sequence Coarsening as an Inductive Bias for Transformers

In addition to the benefits of higher-order signature terms, we empirically observe that even using level-one signature terms resulted in performance improvements when compared to processing sequences without any transformation. We believe that the reduction in input signal length, achieved without significant information loss through the signature transform is another important factor in the improved inductive bias of `RFormer`. This finding aligns with the concurrent work of [4], which highlights some of the drawbacks of decoder-only Transformers for long sequences in terms of both *oversquashing* and *representational collapse*.

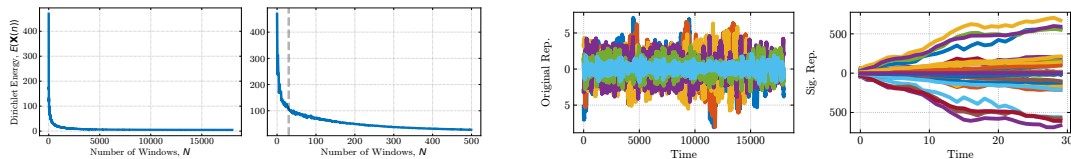

Figure 6: **Left:** Dirichlet energy as a function of window size for the Eigenworms dataset. **Right:** Original and hidden representation after signature layer for two examples in the EW dataset.

To measure the degree of coarsening in the sequence, we find that interpreting the temporal sequence as a path graph and using ideas from the oversmoothing literature [75] serves as a good way to measure the similarity of the representations being fed to the Transformer. In particular, we compute the Dirichlet Energy [74], defined in this case as $E(\mathbf{X}) = \frac{1}{N} \sum_{i=1}^{N} ||\mathbf{X}_i - \mathbf{X}_{i-1}||_2$ of the temporal sequence resulting from taking increasing window sizes of the global signature. An example of this is shown in Figure 6 for the `Eigenworms` dataset, where we compared different numbers of windows (from 2 to 18k). Interestingly, we found that the "elbow" of the Dirichlet energy corresponded to 30 windows in this dataset, which we found empirically to be one of the most performant settings. This hints at the idea of the Dirichlet energy being used for signature hyperparameter tuning as well.

## 6 Conclusion

We introduced the Rough Transformer, a variant of the original Transformer that allows the processing of discrete-time series as continuous-time signals through the use of multi-view signature attention. Empirical comparisons showed that Rough Transformers outperform vanilla Transformers and continuous-time models on a variety of time-series tasks and are robust to the sampling rate of the signal. Finally, we showed that RFormer provides significant speedups in training time compared to regular attention and ODE-based methods, without the need for major architectural modifications or sparsity constraints.

## Impact Statement

This work is unlikely to result in any harmful societal repercussions. Its primary potential lies in its ability to enhance and advance existing data modelling and machine learning methods.

## Acknowledgements

We thank Christopher Salvi, Antonio Orvieto, Yannick Limmer, and Benjamin Walker for discussions at different stages of the project. AA acknowledges support from the Rafael Del Pino Foundation and a G-Research travel grant. XD acknowledges support from the Oxford-Man Institute of Quantitative Finance.

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

# A Properties of Path Signatures

First, we recall that the path is uniquely determined by its signature, which motivates its use as a feature map.

**Proposition A.1.** *Given a path $\widehat{X} : [0, T] \rightarrow \mathbb{R}^d$, then the map $P : [0, T] \rightarrow \mathbb{R}^{1+d}$ where $P(t) = (t, \widehat{X}(t))$ is uniquely determined by it's signature $S(P)_{0,T}$.*

The proof can be found in Hambly and Lyons [33].

For Rough Transformers, several features of path signatures are important. First, linear functionals on path signatures possess universal approximation properties for continuous functionals.

**Theorem A.2.** *Fix $T > 0$, and let $K \subset C_b^1([0, T]; \mathbb{R}^d)$. Let $f : K \rightarrow \mathbb{R}$ be continuous with respect to the sup-norm topology on $C_b^1([0, T]; \mathbb{R}^d)$. Then for any $\epsilon > 0$, there exists a linear functional $\ell$ such that*

$$|f(\overline{X}) - \langle \ell, S(\overline{X})_{0,T} \rangle| \leq \epsilon, \tag{13}$$

*for any $\widehat{X} \in K$, where $\overline{X}$ denotes the time-added augmentation of $\widehat{X}$.*

For a proof of A.2, see Arribas [2]. Even though Theorem A.2 guarantees that *linear* functionals are sufficient for universal approximation, linear models are not always sufficient in practice. This motivates the development of nonlinear models built upon the path signature which efficiently extract path behavior.

The second feature is that the terms of the path signature decay factorially, as described by the following proposition.

**Proposition A.3.** *Given $\widehat{X} \in C_b^1([0, T]; \mathbb{R}^d)$, for any $s, t \in [0, T]$, we have that for any $I \in \mathcal{I}_d^n$*

$$|S(\widehat{X})_{0,T}^I| = O\left(1/n!\right). \tag{14}$$

For a proof of Proposition A.3, see [51]. Hence, the number of terms in the signature grows exponentially in the level of the signature, but the tail of the signature is well-behaved, so only a few levels in a truncated signature are necessary to adequately approximate continuous functionals.

## A.1 Signatures of Piecewise Linear Paths.

In the Rough Transformer, we use linear interpolation of input time-series to get a continuous-time representation of the data. As mentioned in Section 3, the signature computation in this case is particularly simple.

Suppose $\widehat{X}_k : [t_k, t_{k+1}] \rightarrow \mathbb{R}^d$ is a linear interpolation between two points $X_k, X_{k+1} \in \mathbb{R}^d$. That is,

$$\widehat{X}_k(t) = X_k + \frac{t - t_k}{t_{k+1} - t_k}\left(X_{k+1} - X_k\right). \tag{15}$$

Then the signature of $\widehat{X}_k$ is given explicitly by

$$S(\widehat{X}_k)_{t_k, t_{k+1}} = \left(1, X_{k+1} - X_k, \frac{1}{2}(X_{k+1} - X_k)^{\otimes 2}, \frac{1}{3!}(X_{k+1} - X_k)^{\otimes 3}, ..., \frac{1}{n!}(X_{k+1} - X_k)^{\otimes n}, ...\right), \tag{16}$$

where $\otimes$ denotes the tensor product. Let $\widehat{X}_k * \widehat{X}_{k+1}$ denote the *concatenation* of $\widehat{X}_k$ and $\widehat{X}_{k+1}$. That is, $\widehat{X}_k * \widehat{X}_{k+1} : [t_k, t_{k+2}] \rightarrow \mathbb{R}^d$ is given by

$$\widehat{X}_k * \widehat{X}_{k+1}(t) = \begin{cases} \widehat{X}_k(t) & t \in [t_k, t_{k+1}] \\ \widehat{X}_{k+1}(t) & t \in (t_2, t_{k+2}]. \end{cases} \tag{17}$$

The signature of the concatenation $\widehat{X}_k * \widehat{X}_{k+1}$ is given by *Chen's relation*, whose proof is in [51]. To state this result, we first note that $S(\widehat{X})_{s,t}^n$ can be interpreted as an element of the *extended tensor algebra* of $\mathbb{R}^d$:

$$T((\mathbb{R}^d)) = \left\{(a_0, ..., a_n, ...) : a_n \in \mathbb{R}^{d \otimes n}\right\}. \tag{18}$$

**Proposition A.4** (Chen's Relation). *The following identity holds:*

$$S(\widehat{X}_k * \widehat{X}_{k+1})_{t_k, t_{k+2}} = S(\widehat{X}_k)_{t_k, t_{k+1}} \otimes S(\widehat{X}_{k+1})_{t_{k+1}, t_{k+2}}, \tag{19}$$

*where for elements $A, B \in T((\mathbb{R}^d))$ with $A = (A_0, A_1, A_2, ...)$ and $B = (B_0, B_1, B_2, ...)$ the tensor product $\otimes$ is defined*

$$A \otimes B = \left( \sum_{j=0}^{k} A_j \otimes B_{k-j} \right)_{k \geq 0}. \tag{20}$$

Let $\mathbf{X} = (X_0, ..., X_L)$ be a time-series. Then the linear interpolation $\tilde{X} : [0, T] \to \mathbb{R}^d$ can be represented as the concatenation of a finite number of linear paths:

$$\tilde{X} = \widehat{X}_0 * \cdots * \widehat{X}_{L-1}. \tag{21}$$

Hence, the signature is

$$S(\tilde{X})_{0,T} = S(\widehat{X}_0)_{0,t_1} \otimes \cdots \otimes S(\widehat{X}_{L-1})_{t_{L-1}, T}. \tag{22}$$

## B  Related Work, Experimental Choices, and Impact Statement

**Continuous-time models.** Since their introduction in [12], Neural ODEs were extended in various ways to facilitate modelling continuous time-series data [70, 60, 34, 40, 79]. While Neural ODEs and their extensions are successful in certain tasks they are burdened with a high computational cost, which makes them scale very poorly to long sequences in the time-series setting. Various authors propose methods and augmentations to vanilla Neural ODEs to decrease their computational overhead [24, 6]. Other approaches to augmenting deep learning methods to modelling continuous data include implicit neural representations [80, 29], continuous kernel convolutions [69], or Fourier neural operators [50, 63].

**Transformers.** First proposed in [86], the Transformer has been exceptionally successful in discrete sequence modelling tasks such as natural language processing (NLP). Key to the success of the Transformer in NLP is the attention mechanism, which extracts long-range dependencies. There are a number of extensions to improve efficiency and decrease the computation cost of the attention mechanism [49, 88, 22, 41, 16].

**Signatures in machine learning.** The path signature originates from theoretical stochastic analysis [51] and has since become a popular tool in machine learning. Path signatures are regarded as effective feature transformations for sequential data [64, 27, 44]. Additionally, signatures help mitigate the computational cost of Neural CDEs in long time-series [57] and non-Markovian stochastic control problems [39]. Other more recent works in this direction include [18, 87]. Approaches such as randomized signatures [21, 19] and the signature kernel [47, 77] have been developed to mitigate the curse of dimensionality inherent in path signature computations. Rough Transformers provide a first step towards incorporating path signatures for continuous-time sequence modelling using Transformers. [6]

We also note that contemporary work [84] employs a Transformer architecture with signature features for the task of deep hedging. However, our work differs in several key aspects. First, we introduce the multi-view attention mechanism, which uses signatures to extract both global and local information, which we found to be necessary in our experimentation, as Transformers are known to struggle in extracting local information (see Figure 7), whereas their work just uses a global signature. Moreover, their work computes the signature at every time step, strictly dilating input data. This is particularly problematic for long, multi-variate sequences, for reasons discussed above, and can actually negatively impact performance. Our work, however, *compresses* data using the multi-view signature transform, and we find that this compressed representation can actually improve performance. Finally, their work relies on the assumption that data is regularly sampled, as the signature is computed at every time step, in contrast to our work which is robust to irregular sampling.

---

[6]For a preliminary version of this paper, we also direct the reader to [56].

**Long-Range Sequence modelling.** A highly relevant line of research related to enhancing recurrent neural networks' capability to capture long-term dependencies involves the development of various models. These include Unitary RNNs [1], Orthogonal RNNs [38], expRNNs [48], chronoLSTM [82], antisymmetric RNNs [10], Lipschitz RNNs [25], coRNNs [71], unicoRNNs [72], LEMs [73], waveRNN [42], Linear Recurrent Units [62], and Structured State Space Models [32, 31]. While we utilize many benchmarks and synthetic tasks from these works to test our model, it is important to note that our work is not intended to compete with the state-of-the-art in these tasks. Therefore, we do not directly compare our model with the models mentioned above. Instead, this paper seeks to show that the baseline Transformer architecture can benefit from the use of signatures by (i) becoming more computationally efficient, (ii) being invariant to the sampling rate of the signal, and (iii) having a good inductive bias for temporal and spatial processing. Furthermore, we highlight that `RFormer` brings alternative benefits, such as the ability to perform spatial processing effectively, which is a setting in which long-range sequence models typically struggle.

**Efficient Attention Variants.** There are several efficient self-attention variants that have emerged over the years, including Sparse Transformer [14], Longformer [5], Linear Transformers [41], BigBird [90], Performer [16], or Diffuser [26]. In our setting, we highlight that a central part of this paper is to showcase how signatures significantly reduce the computational requirements of vanilla attention and empirically demonstrate that this also results in improved learning dynamics and invariance to the sampling frequency of the signal. Given the large efficiency gains that we observed with this approach when employed on vanilla attention, we did not consider that further experimentation on other forms of "approximate" attention was needed. Since most variants of attention seek to make the operation more efficient through several approximations (e.g., linearization or sparsification techniques), we believe that a first attempt at showcasing the power of multi-view signatures on vanilla attention is already significant. However, other variants of attention (such as the ones outlined before) could be added on top of the signature representations to obtain even better efficiency gains.

**Limitations and Future Work.** While we found RFormer to be very performant in our experiments, much of this performance gain relies on heavy hyperparameter tuning, especially when it comes to the choice of window sizes and signature level. However, this could be handled using Neural Architecture Search (NAS) techniques, such as those employed in [76]. Furthermore, despite the computational gains we achieve for low-dimensional sequences, additional work would be required to scale this method to larger dimensions. We should also note that the experiments and results presented in this paper are constrained by the relatively small scale of the models studied.

## C  Experimental Details

All experiments are conducted on an NVIDIA GeForce RTX 3090 GPU with 24,564 MiB of memory, utilizing CUDA version 12.3. Hyperparameters used to produce the results in Table 2 are reported in Tables 6. The timings presented in all tables are obtained by executing each model independently for each dataset and averaging the resulting times across 100 epochs.

Table 6: Hyperparameters used for Table 2, where G and L refer to the Global and Local signature components, respectively.

|  | SCP1 | SCP2 | MI | EW | ETC | HB |
|---|---|---|---|---|---|---|
| Batch Size | 20 | 10 | 50 | 5 | 10 | 20 |
| Embedded Dim | 10 | 5 | 20 | 20 | 20 | 5 |
| Multi-View Terms | [G] | [G] | [L] | [L] | [G] | [G, L] |
| Learning Rate | 4.08e-3 | 1.38e-3 | 4.08e-3 | 6.73e-3 | 1.00e-3 | 7.72e-3 |
| Num. Heads | 3 | 3 | 3 | 1 | 1 | 3 |
| Num. Layers | 2 | 3 | 3 | 2 | 1 | 3 |
| Num. Sig Windows | 100 | 50 | 200 | 10 | 400 | 30 |
| Sig Level | 2 | 3 | 2 | 2 | 1 | 2 |
| Univariate | true | true | true | false | false | true |
| Num. Epoch | 110 | 10 | 26 | 39 | 200 | 16 |

Table 7: Hyperparameters validation on remaining datasets.

| Dataset | Learning Rate | Number of Windows | Sig. Depth | Sig. Type | Univariate/Multivariate Sig. |
|---|---|---|---|---|---|
| Sinusoidal | $1 \times 10^{-3}$ | 75 | 6 | Multi-View | - |
| HR | $1 \times 10^{-3}$ | 75 | 4 | Local | Multivariate |

To prevent excessive growth in signature terms, we use the univariate signature in LOB datasets. As an alternative, one could employ randomized signatures [19] or low-rank approximations [9, 11] .

## D    Baselines Validation

This section collects the validation of Step and Depth for the Neural-RDE model. Optimal values are selected for evaluation on test-set. Early-stopping is used with the same criteria as [57].

Table 8: Validation accuracy on the sinusoidal dataset.

| Acc. Val | Step | Depth | Memory Usage (Mb) | Elapsed Time (s) |
|---|---|---|---|---|
| 17.26 | 2 | 2 | 778.9 | 6912.7 |
| 12.21 | 2 | 3 | 770.3 | 1194.43 |
| 16.35 | 4 | 2 | 382.2 | 2702.48 |
| 19.27 | 4 | 3 | 386.16 | 574.97 |
| 20.99 | 8 | 2 | 193 | 1321.36 |
| **24.02** | **8** | **3** | **194.17** | **332.17** |
| 17.15 | 16 | 2 | 97.13 | 136.43 |
| 21.59 | 16 | 3 | 98.17 | 156.93 |
| 17.46 | 24 | 2 | 65.96 | 105.94 |
| 20.59 | 24 | 3 | 66.68 | 98.97 |

Table 9: Validation accuracy on the long sinusoidal dataset.

| Acc. Val | Step | Depth | Memory Usage (Mb) | Elapsed Time (s) |
|---|---|---|---|---|
| 11.10 | 2 | 2 | 4017.22 | 2961.98 |
| 9.59 | 2 | 3 | 4008.33 | 2779.52 |
| 10.39 | 4 | 2 | 2001.76 | 1677.78 |
| 10.19 | 4 | 3 | 2006.80 | 1615.64 |
| 14.03 | 8 | 2 | 1004.07 | 665.55 |
| **15.34** | **8** | **3** | **1005.72** | **723.41** |
| 1.61 | 16 | 2 | 503.66 | 125.85 |
| 1.92 | 16 | 3 | 505.28 | 120.63 |
| 1.51 | 24 | 2 | 339.80 | 58.87 |
| 2.12 | 24 | 3 | 341.90 | 69.35 |

Table 10: Validation accuracy on the EW dataset.

| Acc. Val | Step | Depth | Memory Usage (Mb) | Elapsed Time (s) |
|---|---|---|---|---|
| 84.62 | 2 | 2 | 5799.40 | 21289.99 |
| **87.18** | **2** | **3** | **6484.93** | **25925.80** |
| 79.49 | 4 | 2 | 2891.61 | 11449.14 |
| 82.05 | 4 | 3 | 3240.99 | 9055.12 |
| 82.05 | 8 | 2 | 1446.94 | 4143.26 |
| 76.92 | 8 | 3 | 1624.73 | 3616.43 |
| 82.05 | 16 | 2 | 724.35 | 1909.69 |
| 76.92 | 16 | 3 | 817.04 | 1924.27 |
| 79.49 | 24 | 2 | 483.92 | 1098.21 |
| 74.36 | 24 | 3 | 543.78 | 987.02 |

Table 11: Validation loss on the HR dataset.

| Acc. Val | Step | Depth | Memory Usage (Mb) | Elapsed Time (s) |
|---|---|---|---|---|
| **2.44** | **2** | **2** | **5044.44** | **56492.33** |
| 3.03 | 2 | 3 | 5059.28 | 39855.19 |
| 3.67 | 4 | 2 | 2515.40 | 10765.58 |
| 16.04 | 4 | 3 | 2531.44 | 7157.20 |
| 5.35 | 8 | 2 | 1259.30 | 3723.94 |
| 2.70 | 8 | 3 | 1268.60 | 18682.82 |
| 3.58 | 16 | 2 | 632.08 | 3518.96 |
| 3.64 | 16 | 3 | 636.64 | 7922.96 |
| 3.86 | 24 | 2 | 422.74 | 3710.95 |
| 3.55 | 24 | 3 | 426.83 | 6567.02 |

Table 12: Validation loss on the LOB dataset (1K), included as an additional experiment in Appendix **??**.

| Val Loss | Step | Depth | Memory Usage (Mb) | Elapsed Time (s) |
|---|---|---|---|---|
| **0.58** | **2** | **2** | **1253.55** | **180.79** |
| 1.74 | 2 | 3 | 1447.57 | 308.52 |
| 1.58 | 4 | 2 | 623.87 | 71.18 |
| 32.90 | 4 | 3 | 754.05 | 87.81 |
| 2.94 | 8 | 2 | 317.40 | 61.27 |
| 4.84 | 8 | 3 | 406.88 | 62.71 |
| 2.24 | 16 | 2 | 164.70 | 18.67 |
| 6.26 | 16 | 3 | 234.92 | 24.20 |
| 3.82 | 24 | 2 | 112.80 | 12.69 |
| 15.35 | 24 | 3 | 176.68 | 14.92 |

# E   Long Temporal Datasets Details

Table 13 summarises the long temporal modeling datasets from the UEA time series classification archive [3] used in Section 4.

Table 13: Summary of datasets used in the long time-series classification task.

| Dataset | #Sequences | Length | #Classes | #Dimensions |
|---|---|---|---|---|
| SelfRegulationSCP1 (SCP1) | 561 | 896 | 2 | 6 |
| SelfRegulationSCP2 (SCP2) | 380 | 1152 | 2 | 7 |
| MotorImagery (MI) | 378 | 3000 | 2 | 64 |
| EigenWorms (EW) | 259 | 17984 | 5 | 6 |
| EthanolConcentration (ETC) | 524 | 1751 | 4 | 3 |

# F  Ablation Studies

## F.1  Global and Local Signature Components

In this section, we ablate the use of the multi-view signature transform over both global and local transformations of the input signal. The results for the sinusoidal datasets are shown in Figure 7. In most cases, the use of both local and global components improves the performance of `RFormer`. This choice, however, can be seen as a hyperparameter and will be dataset-dependent.

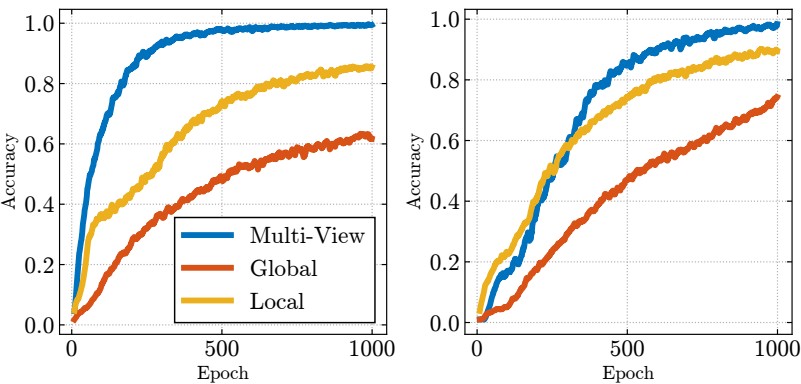

Figure 7: Ablation of local and local components of the multi-view signature for the sinusoidal datasets. **Left:** Sinusoidal dataset. **Right:** Long Sinusoidal dataset.

## F.2  Signature Level and Naive Downsampling

One of the main points of the paper is that the shorter representation of the time-series endowed by the signatures helps to significantly reduce the computational cost of the self-attention operation with minimal information loss (and with improved performance in many of the experiments). By equation (16), one sees that the first level of the signature of a linear function is the difference between its endpoints. Hence, using multi-view attention with signature level one operates on the increments of piecewise-linear interpolated data, which corresponds to naive downsampling. To test that higher levels of the signature provide improvements in performance, we compare the result of using the signature on the datasets tested in Table 14 below.

Table 14: Comparative performance of different methods on datasets.

| Dataset | Linear-Interpolation + Vanilla | Rough Transformer with sig level (n) | Improvement |
|---|---|---|---|
| EigenWorms | 64.10% | 90.24% (2) | 40.77% |
| HR | 10.56 | 2.66 (4) | 74.81% |

There is a significant performance gain in considering higher levels of the signature because one can capture the higher-order interactions between the different time-series.

# G   Additional Experiments and Comparisons

## G.1   Random Drop Experiments

Furthermore, we conduct a new set of experiments in which we dropped 30% and 70% of the dataset for RFormer. Note that even with a 70% drop rate in the EigenWorms dataset, the vanilla Transformer fails to run due to memory limitations. Therefore, to provide results for the Transformer model on the EigenWorms dataset, we conduct experiments with an 85% drop rate. This comparison highlights the performance gap between the vanilla Transformer and our proposed model under these conditions, with the RFormer model yielding superior results. All results are computed across five seeds and are summarized in the tables and figure below.

Table 15: Performance of models under various data drop scenarios for EW dataset.

| Model | Full | 30% Drop | 50% Drop | 70% Drop | 85% Drop |
|---|---|---|---|---|---|
| Transformer | OOM | OOM | OOM | OOM | $72.45\% \pm 3.36$ |
| RFormer | $90.24\% \pm 2.15$ | $87.86\% \pm 3.28$ | $87.69\% \pm 4.97$ | $83.35\% \pm 2.86$ | $82.74\% \pm 2.13$ |

Table 16: Performance consistency of RFormer under data drop scenarios for HR dataset.

| Model | Full | 30% Drop | 50% Drop | 70% Drop |
|---|---|---|---|---|
| RFormer | $2.66 \pm 0.21$ | $2.72 \pm 0.19$ | $2.82 \pm 0.05$ | $2.98 \pm 0.08$ |

Table 17: Epoch-wise performance under different data drop scenarios for the sinusoidal dataset.

| | Epoch 100 | Epoch 250 | Epoch 500 | Epoch 1000 |
|---|---|---|---|---|
| 30% Drop | 48.6% | 82.5% | 91.4% | 99.3% |
| 70% Drop | 35.7% | 56.8% | 64.9% | 67.8% |

Table 18: Epoch-wise performance under different data drop scenarios for the long sinusoidal dataset.

| | Epoch 100 | Epoch 250 | Epoch 500 | Epoch 1000 |
|---|---|---|---|---|
| 30% Drop | 39.1% | 72.6% | 96.2% | 98.2% |
| 70% Drop | 27.5% | 66.7% | 78.5% | 85.3% |

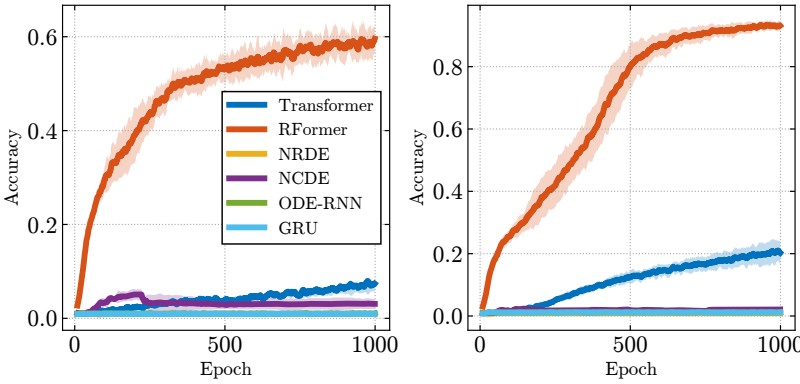

Figure 8: Test accuracy per epoch for the frequency classification task across three random seeds for sinusoidal datasets with 50% random drop per epoch. **Left:** Sinusoidal dataset. **Right:** Long Sinusoidal dataset.

Finally, Table 19 compares CRU and RFormer in an irregularly sampled synthetic data setting, featuring shorter sinusoids and fewer classes than the experiments in Section 4.1. Additionally, Table 20 presents the hyperparameter validation for CRU (see Table 21 for training time analysis). These experiments demonstrate that recurrent models perform well with short sequences. Note that despite RFormer's superior performance, our model is significantly faster than other continuous-time models, as shown in Appendix G.2, particularly in Table 21.

Table 19: Comparison of RFormer and CRU (two best and simplest performing instances [Num.basis/Bandwidth= 20/3]) at different random drop percentages.

| L | Random Drop | RFormer | CRU (LSD=10) | CRU (LSD=20) |
|---|---|---|---|---|
| | 0% | 100.00% | 100% | 100.00% |
| | 30% | 98.60% | 65.90% | 99.60% |
| 100 | 50% | 97.80% | 34.40% | 94.70% |
| | 70% | 96.10% | 43.00% | 78.60% |
| | 85% | 85.50% | 32.30% | 57.30% |
| | 0% | 100.00% | 100.00% | 100% |
| | 30% | 99.90% | 42.95% | 94.90% |
| 250 | 50% | 99.40% | 43.65% | 77.30% |
| | 70% | 98.30% | 45.40% | 94.40% |
| | 85% | 86.20% | 38.80% | 83.60% |
| | 0% | 100.00% | 100.00% | OOM |
| | 30% | 99.90% | 47.15% | OOM |
| 500 | 50% | 99.70% | 48.80% | OOM |
| | 70% | 99.30% | 55.15% | OOM |
| | 85% | 87.70% | 46.50% | OOM |

Table 20: CRU's hyperparameters ($L = 100$) (latent state dimension (LSD), number of basis matrices (Num.basis), and their bandwidth).

| LSD | Num. basis | Bandwidth | Acc (30 Epochs) |
|---|---|---|---|
| | 15 | 3 | 78% |
| 10 | 15 | 10 | - |
| | 20 | 3 | 100% |
| | 20 | 10 | - |
| | 15 | 3 | 81.30% |
| 20 | 15 | 10 | 91.70% |
| | 20 | 3 | 100% |
| | 20 | 10 | 99.90% |
| | 15 | 3 | 99.90% |
| 40 | 15 | 10 | 97.50% |
| | 20 | 3 | 100% |
| | 20 | 10 | 100% |

## G.2 Additional Efficiency Experiments and Discussion

We conduct additional experiments to compare the runtime of Rough Transformers with other models. In this experiment, we use the synthetic sinusoidal dataset considered in our paper and compute the runtime per epoch for varying sequence lengths. We demonstrate results for two variants of RFormer: "online", which corresponds to computing the signatures of each batch during training (resulting in significant redundant computation), and "offline", which corresponds to computing the signatures in one go at the beginning of training. We include a recent RNN-based model as a basis for comparison with high-performing RNN baselines. In addition to the models discussed in Section 4, we introduce Continuous Recurrent Units (CRU) [78] as a new baseline. See Table 21 for a summary of the results.

Table 21: Seconds per epoch for growing input length and for different model types on the sinusoidal dataset.

| Model | S/E for Varying Context Length ↓ | | | | | | | |
|---|---|---|---|---|---|---|---|---|
| | **L=100** | **L=250** | **L=500** | **L=1000** | **L=2500** | **L=5000** | **L=7.5k** | **L=10k** |
| NRDE | 5.87 | 11.67 | 20.27 | 44.01 | 103.11 | 201.21 | 312.31 | 467.47 |
| NCDE | 42.59 | 121.82 | 225.14 | 458.09 | 1126.77 | 2813.42 | 4199.50 | 5345.39 |
| GRU | 1.56 | 1.55 | 1.65 | 1.63 | 1.78 | 2.37 | 3.65 | 4.79 |
| CRU | 59.22 | 199.15 | 789.28 | OOM | OOM | OOM | OOM | OOM |
| ContiFormer | 61.36 | 248.31 | 1165.02 | OOM | OOM | OOM | OOM | OOM |
| Transformer | 0.75 | 0.79 | 0.82 | 0.95 | 1.36 | 5.31 | 9.32 | 16.32 |
| RFormer (Online) | 0.75 | 0.88 | 0.94 | 1.03 | 1.28 | 1.55 | 1.83 | 2.35 |
| RFormer (Offline) | 0.67 | 0.64 | 0.63 | 0.65 | 0.60 | 0.59 | 0.62 | 0.60 |

We remark that previous running times are obtained with a batch size of 10. Further, the `ContiFormer` model could be run for $L = 1000$ if decreasing the batch size to 2 (which significantly affects the parallelization process), avoiding OOM issues and resulting in 4025 seconds/epoch, which is several orders of magnitude larger than `RFormer`. As an additional experiment, we tested the epoch time (S/E) of `RFormer` for extremely oversampled sinusoidal time series. We show our results in the table below.

Table 22: Seconds per epoch for very large input length.

| Model | S/E for Varying Context Length ↓ | | | |
|---|---|---|---|---|
| | **L=25k** | **L=50k** | **L=100k** | **L=250k** |
| RFormer (Online) | 5.39 | 9.06 | 19.95 | 45.20 |
| RFormer (Offline) | 0.60 | 0.61 | 0.60 | 0.63 |

Thus, the time needed to compute the signature is inconsequential when compared with the time required to train standard models on the full or even downsampled datasets, since this step has to be carried out only once. To put this into context with an example, we note that it takes 4s to compute the signature representations for the HR dataset (which is about half the time it takes for the Vanilla Transformer to go through one epoch) and results in a 26× increase in computational speed for RFormer when compared to the vanilla Transformer.

Table 23: Processing times for different sizes on the sinusoidal dataset.

| Size | 100 | 250 | 500 | 1k | 2.5k | 5k | 7.5k | 10k | 25k | 50k | 75k | 100k |
|---|---|---|---|---|---|---|---|---|---|---|---|---|
| **Time** | 0.15 s | 0.21 s | 0.24 s | 0.39 s | 0.42 s | 0.51 s | 0.70 s | 1.09 s | 1.64 s | 2.94 s | 4.49 s | 5.74 s |

To showcase that this is the case for not only sequences of moderate length but also extremely long sequences, we also carry out the following experiment where we compute the signature representation for the sine dataset, with a progressively increasing number of datapoints. As seen in Table 23, this does not cause an explosion in computational time.

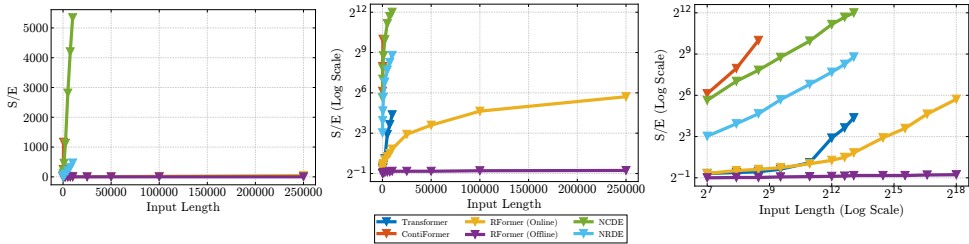

Figure 9: Seconds per epoch for growing input length and for different model types on the sinusoidal dataset for extremely long lengths (up to 250k) **Left:** Log Scale. **Middle:** Regular Scale. **Right:** Log-log scale. When a line stops, it indicates an OOM error.

### G.3 Additional ContiFormer Comparisons

Also, to provide some context of the performance of `ContiFormer` compared with our method (and not only results on complexity and training times), we run the model on the sinusoidal classification task for signals of length $L = 100$ and $L = 250$. Due to the slow running time of the `ContiFormer` model, we did not consider sequence lengths of $L > 250$. We evaluate the `ContiFormer` model using one head. However, given the subpar results we obtain, we also test it with four heads, using the hyperparameters originally used in the paper for their irregularly sampled time series classification experiments. By contrast, all variations of `RFormer` tested in this paper for this experiment employ only one head, but reported significantly better results.

Table 24: Model performance for $L = 100$.

| Model | Epoch 100 | Epoch 250 | Epoch 500 |
|---|---|---|---|
| ContiFormer (1 Head) | 2.3% | 2.8% | 3.1% |
| ContiFormer (4 Heads) | 8.5% | 17.3% | 20.0% |
| Transformer (1 Head) | 13.7% | 40.1% | 82.8% |
| RFormer (1 Head) | 38.7% | 81.1% | 92.3% |

