# OpenReview forum: "Rough Transformers: Lightweight and Continuous Time Series Modelling through Signature Patching"
_NeurIPS.cc/2024/Conference — NeurIPS 2024 poster_

### Official Review · Reviewer_hdr3 · 2024-07-09

**Soundness:** 2
**Presentation:** 2
**Contribution:** 2
**Rating:** 4
**Confidence:** 3

**Summary:**

This paper introduces the Rough Transformer, a variant of the original Transformer that allows the processing of discrete-time series as continuous-time signals through the use of multi-view signature attention. Empirical comparisons shows that Rough Transformers outperform vanilla Transformers and continuous-time models on a variety of time-series tasks and are robust to the sampling rate of the signal.

**Strengths:**

- Overall the paper is well written and is easy to follow
- The idea of using signature transform within an attention mechanism is interesting

**Weaknesses:**

- Empirical evaluations are limited, casting doubts on the true potential of the proposed architecture
- The tasks considered are rather simple, and it is not clear whether the proposed architecture will give favorable tradeoffs between accuracy and efficiency in the more challenging tasks (see below)
- Missing evaluations on time series forecasting tasks (only classification and regression tasks are considered)
- Missing comparisons with recent RNN models (such as https://arxiv.org/abs/2110.04744, https://arxiv.org/abs/2212.00228), Transformer models (e.g., those studied in https://arxiv.org/abs/2011.04006), State Space models (https://arxiv.org/abs/2111.00396 and the more recent variants such as Mamba: https://arxiv.org/abs/2312.00752) and other sequence models (https://arxiv.org/abs/2305.01638, https://arxiv.org/abs/2209.10655)
- Missing ablation studies on the components of the proposed architecture, particularly the role of the global and local components in the multi-view signature, truncation level n, etc.
- Missing related work; e.g., the papers mentioned above, https://link.springer.com/article/10.1007/s40304-017-0103-z and https://arxiv.org/abs/1710.10121 for continuous-time DL models, https://arxiv.org/abs/2006.12070, https://arxiv.org/abs/2102.04877 for continuous-time RNN models, https://dl.acm.org/doi/abs/10.5555/3546258.3546305 for using path signatures to understand continuous-time RNNs

**Questions:**

- I was wondering how effective is the proposed model in autoregressive generative tasks and time series forecasting (see https://arxiv.org/abs/2311.04147)? These are natural tasks in the domain of sequence modeling
- While the proposed method can improve the modified transformer model in sequence modeling, why would the method be attractive for practitioners when they could just use more sophisticated models like SSMs for sequence modeling?

**Limitations:**

Yes

---

> ### Author Rebuttal · Authors · 2024-08-06
>
> We thank the reviewer for the feedback. We are slightly surprised by some of their comments, which we believe are already addressed in the manuscript. We hope that this response will help to clarify any misunderstandings. In line with the reviewer's comments, we have carried out an extensive set of new experiments which we believe address all of the reviewer's concerns. We hope the reviewer will take this into account and consider raising the score accordingly.
>
> **[W1, W4] Empirical evaluations are limited / Missing comparisons with recent models.**
>
> In line with the reviewer's feedback, we have added comparisons to LRU, S5, and MAMBA on 5 long temporal modeling new datasets, where we find RFormer to perform very competitively. These results are included in Table 5 of the attached PDF to this rebuttal.
>
> The reason we focus on these tasks is because our method is tailored to time-series processing, and other image or text-based benchmark datasets (such as LRA/sMNIST, etc), which typically test model memory, are not as suitable in this setting. We will add a comment along these lines in the paper as well.
>
> In terms of performance on random dropping, we have added experiments on 15 new datasets, comparing against SoTA models for continuous time series processing. We report the performance and average rank of our model in Table 4 (PDF), where our model consistently outperforms the baselines. We find that our model performs very competitively, as well as being orders of magnitude faster than the baselines.
>
> In terms of the small empirical evaluation, we would like to stress that we have now tested our model on a wide variety of datasets. We  have also run additional experiments whose results can be found in Tables 1, 2, and 3 of the PDF comparing RFormer with CRU [1]. Additionally, Table 4 (PDF) reports the performance of 8 extra baselines on 15 new datasets with different random dropping rates. We consider this extensive evaluation very much in line with the experiments carried out by other sequence modeling architectures  (e.g. NCDE, NRDE, coRNN, noisy RNN). In terms of the tasks considered, we also find that they are consistent with the experiments in these papers, which focused on classification and regression tasks. Nonetheless, we have extended our results into the forecasting scenario through a step-ahead forecasting task, see response to [W2,W3] and the General Response.
>
> **[W5] Missing ablation studies on the components of the proposed architecture, particularly the role of the global and local components in the multi-view signature, truncation level \(n\), etc.**
>
> We would like to clarify that these results are already in the paper. In particular, they can be found in Appendix E: Ablation Studies. We have also carried out a set of experiments on the sensitivity of the model to hyperparameters, which can be found in Figure 2 of the associated PDF document.
>
> **[W6] Missing related work.**
>
> We would like to clarify that we have a dedicated related work section in Appendix B, where we have already included citations to many of the works that the reviewer mentions. In particular, we include a discussion of Unitary RNNs, Orthogonal RNNs, expRNNs, chronoLSTM, antisymmetric RNNs, Lipschitz RNNs, coRNNs, unicoRNNs, LEMs, waveRNN, Linear Recurrent Units, and Structured State Space Models (S4 and Mamba) on lines L678 - L690. Furthermore, we discuss efficient attention variants such as Sparse Transformer, Longformer, Linear Transformers, BigBird, Performer, and Diffuser in lines L691 - L702. We will add the remaining citations the reviewer suggests.
>
> **[W2, W3] The tasks considered are rather simple / Missing evaluations on time series forecasting tasks.**
>
> We would like to echo our previous statement on how similar models (NCDE, NRDE, coRNN, noisy RNN) have tested exclusively in classification and regression tasks and how these models have become widely accepted by the community.
>
> In terms of forecasting, we would like to highlight that time series forecasting pipelines using Transformers typically train by masking the input representation iteratively to predict the next time step. In our setting, since we are compressing the input representation, these pipelines cannot be straightforwardly implemented for our model. We intend to work towards a model that uses these principles just for time series forecasting, but note this would require a completely new training pipeline as well as heavy tuning. That said, we reiterate that the focus of this work is to showcase the benefits of using Rough Path Signatures within the attention mechanism for efficient and continuous time-series processing, as demonstrated through the widely accepted experimental settings of time-series classification and regression.
>
> We have, however, included a step-ahead forecasting task on the Apple Limit Order Book volatility to extend our results into te forecasting scenario. The results can be found in the General Response.
>
> **[Q2] While the proposed method can improve the modified transformer model in sequence modeling, why would the method be attractive for practitioners when they could use more sophisticated models like SSMs for sequence modeling?**
>
> We thank the reviewer for raising this point.
>
> We believe that showing the significant benefits of RFormer when compared to the traditional Transformer is beneficial to the community. There seems to be a wider adoption of hybrid models (SSM + Transformer) by the community given their superior performance [2]. In tasks that require temporal processing, the use of RFormer could potentially make these models have improved inductive biases, allow for continuous-time processing, and increase efficiency and inference speed.
>
> [1] Schirmer, Mona, et al. "Modeling irregular time series with continuous recurrent units." International conference on machine learning. 2022.
>
> [2] Lieber, Opher, et al. "Jamba: A hybrid transformer-mamba language model." arXiv preprint (2024).

---

### Official Review · Reviewer_tNoo · 2024-07-13

**Soundness:** 3
**Presentation:** 3
**Contribution:** 2
**Rating:** 6
**Confidence:** 4

**Summary:**

The paper proposes Rough Transformers, an attention-based model for long continuous-time signals. The model utilizes ideas from rough path theory to extract path signatures from the continuous-time signal (obtained by interpolation of the original signal). Two types of signatures are extracted: global and local. The global signature extracts long-term information from the signal whereas the local signature extracts local information. Self-attention is then used on this "multi-view" signature. Experiments on classification and regression tasks show improved performance over existing model architectures.

**Strengths:**

- The paper views the problem of modeling long continuous-time signals through the lens of rough path theory. While this in itself is not novel, the combination of rough path signatures with attention is a novel combination.
- Experiments on classification and regression tasks show that RFormer improves over existing models in terms of accuracy and significantly more compute efficient. (I have some concerns and questions about the empirical analysis, please see Weaknesses)

**Weaknesses:**

- The technical contribution is limited. In such scenarios, the empirical analysis needs to be sufficiently strong.
- The empirical analysis has been conducted on a few toyish datasets. While the results are definitely promising, more experimental support is needed to validate the model.
    - Although the model is motivated from a continuous-time and irregularly-sampled data perspective, the actual investigation of these settings is limited. As per my understanding, all experiments under 4.1 have been conducted on regularly sampled time series (please correct me, if I am wrong). I find it surprising that simple RNN-based methods do not perform well in these settings. If this is indeed the case, some simple CNN-based method should be studied. Recent models based on state space layers (e.g., S4, Mamba) can also be explored as baselines.
   - OOM for important baselines is not really helpful to draw any conclusions. To highlight efficiency of RFormer, please conducted experiments where you increased the context length or other parameters to show where baselines run OOM and how do they perform before that.
   - More experiments are needed, particularly for the forecasting task to understand how well the model understands the dynamics.
   - I took a brief look at the code and it looks like hyperparameter tuning was conducted for RFormer. Was such tuning also performed for the baselines? If no (which is a valid response), how did you selected the baseline parameters? How sensitive is the model to different hyperparameters?
- The discussion on related work needs to be moved to the main text and improved. Please contrast RFormer with the related works, particularly the ones that are closely related such as NRDE and ContiFormer. Discussion on some closely related works [1, 2, 3] is missing. Ideally there should also be a comparison with at least of these methods (e.g., CRU).

[1] Schirmer, Mona, et al. "Modeling irregular time series with continuous recurrent units." International conference on machine learning. PMLR, 2022.
[2] Ansari, Abdul Fatir, et al. "Neural continuous-discrete state space models for irregularly-sampled time series." International Conference on Machine Learning. PMLR, 2023.
[3] Oh, YongKyung, Dongyoung Lim, and Sungil Kim. "Stable Neural Stochastic Differential Equations in Analyzing Irregular Time Series Data." arXiv preprint arXiv:2402.14989 (2024).

I am happy to update my score, if my concerns are adequately addressed.

**Questions:**

- Can the authors clarify by what they mean by "input sequences must be sampled at the same times, (ii) the sequence length must be fixed"? These do not seem to be limitations of transformers, especially (ii).

See weaknesses for other questions.

**Limitations:**

There is a brief discussion on limitations.

---

> ### Author Rebuttal · Authors · 2024-08-06
>
> We thank the Reviewer for the engaging review and valuable feedback. We believe we have incorporated the reviewer's suggestions into our manuscript and hope our changes warrant an increase in the score.
>
> **[W1] Need for stronger empirical analysis.**
>
> We have added 19 new datasets, 8 extra baselines, and new synthetic experiments (Tables 4-5/1-3 in the PDF). We hope these additions address the reviewer's concerns and provide a stronger empirical analysis.
>
> **[W2.1] Investigation of continuous-time and irregularly-sampled data limited.**
>
> Most tasks are indeed regularly sampled, except for the Limit Order Book data. However, we performed random drop experiments in Section 4.3 and Appendix F, expanding Section 4.1 to irregularly sampled settings.
>
> We conducted additional experiments to enhance our model's empirical trustworthiness. For regularly sampled data, our model was tested against SOTA models (S5, Mamba, LRU) across 5 datasets, showing very competitive performance (Table 5 PDF).
>
> For irregularly sampled settings, we compared our model against 4 additional baselines, including [1,3] as suggested by the reviewer, across 15 datasets from the UCR time-series archive. Table 4 (PDF) shows our model's performance and average rank (due to space constraints), consistently outperforming the baselines.
>
> Further, Table 2 (PDF) compares CRU and RFormer in an irregularly sampled synthetic data setting (with shorter sinusoids and a smaller number of classes than the experiments in Section 4.1), supported by CRU's hyperparameters validation in Table 1 and training time analysis in Table 3. These experiments demonstrate that recurrent models perform well with short sequences. Note that despite its superior performance, our model is significantly faster than other continuous-time models, as shown in Appendix F.2 and Table 3 (PDF).
>
> **[W2.2] OOM results and efficiency plots.**
>
> We thank the reviewer for raising this point. Regarding OOM errors for ContiFormer, Appendix E of the ContiFormer paper demonstrates its memory requirements scale exponentially with sequence length, making it impossible to run on long time-series datasets considered in our paper. We felt this is worth noting because we think ContiFormer is most similar in spirit to RFormer in that it augments the attention mechanism in a continuous-time fashion. We hope you find that with the additional baselines, this point is made clearer.
>
> We want to highlight that efficiency experiments can be already found in the paper in Figure 4 (Section 4.2. Training efficiency), showing seconds per epoch of all models. When a line stops,  it indicates an OOM error. Appendix F.2 contains more information on computational times, showing RFormer can be used on context lengths of 250k time steps. We apologize if this was not clear and will modify the text accordingly.
>
> **[W2.3] More experiments, particularly for forecasting.**
>
> We have significantly increased the number of experiments during this rebuttal, adding 19 new datasets and 8 new baselines (see responses to [W1,2.1]) and new synthetic experiments.
>
> Regarding forecasting, we should remark that forecasting pipelines using Transformers typically involve iteratively masking the input representation to predict the next time step. In our setting, since we compress the input representation, these pipelines cannot be directly applied and would require extensive modifications, which is not feasible during the rebuttal timeframe. However, we include a step-ahead forecasting of the Apple Limit Order Book volatility (see General Response). Further, we point the reviewer to the HR task, which contains classical temporal dynamics (e.g. cyclicality).
>
> We would like to highlight that the main points of our paper on efficiency, inductive bias, and continuous processing should hold with our experiments. Furthermore, we would like to point out that other models in the area (such as [1,3]) have also been applied in the same tasks and have not been tested for forecasting, which requires extensive tuning.
>
> **[W2.4] Hyperparameter tuning for baselines.**
>
> In short, yes, we did. We thoroughly validated the step and depth parameters used to compute the log-signatures for the NRDE model (Appendix D). We did this as these are the only hyperparameters for which we performed tuning in our model.
>
> For the ODE-RNN and NCDE/NRDE models, we validated the architectural choices. Due to occasional sub-optimal performance in our replications, we included the original results for shared datasets (see Tables 1,3). For ContiFormer, we used hyperparameters from the official repository, testing with both 1 and 4 heads (our model uses 1 head), as detailed in Appendix F.3. For the new datasets, we used optimal hyperparameters from manuscripts/repositories, except for CRU, which we validated ourselves.
>
> Regarding the sensitivity of the model to hyperparameters, we direct the reviewer to Appendix E and have included new sensitivity experiments in Figure 2 of the PDF.
>
> **[W3] Improving related work section.**
>
> For the final version, we will move the discussion on related work to the main text as suggested and we will contrast our work with other models, highlighting (i) the recurrent vs. attention-based structure, (ii) computational efficiency, and (iii) the treatment of the signature with local vs. global windows in the case of NRDEs.
>
> **[Q1] Clarifications.**
>
> What we mean here is that the input data should be regularly sampled (to retain decent performance), and each input sequence must be sampled the same number of times. In many practical scenarios (e.g., medical data), time-series may vary wildly in their length. For Transformers, which have a fixed context window, this poses a non-trivial challenge of how to standardize the input data such that it may be encoded by the model. As suggested by reviewer wNkv, this can be improved through padding. We will add some discussion along these lines and tone down some of our wording.

---

> > ### Comment · Reviewer_tNoo · 2024-08-12
> >
> > Thank you for your thorough responses. Many of my concerns have been addressed, so I am raising my score to 6. Looking forward to the final version of the paper. :)

---

> > > ### Author Response · Authors · 2024-08-12
> > > **Reply to Reviewer**
> > >
> > > Thank you again for your thorough and constructive review, recommending acceptance, and raising the score!

---

### Official Review · Reviewer_wNkv · 2024-07-29

**Soundness:** 2
**Presentation:** 3
**Contribution:** 3
**Rating:** 7
**Confidence:** 4

**Summary:**

The paper proposes Rough Transformer (RFormer), an extention of the Transformer architecture towards operating on continuous-time representations of time-series data. RFormer employs a novel technique called multi-view signature attention, which performs attention on path signatures pre-computed from input data offline, thereby capturing both local and global dependencies across observations. Experiments on various real-world time-series datasets shows that RFormer enjoys superior predictive performance as well as computational efficiency compared to previous methods, while being robust to changes in sequence length and irregular sampling.

**Strengths:**

- [S1] **Good novelty.** To the best of my knowledge, incorporating rough path theory to time-series representation learning is a novel approach, and would be of great interest to the machine learning community.

- [S2] **Great empirical performance.** Experiments on a wide variety of real-world datasets show large improvements in both accuracy and efficiency, demonstrating strong utility of proposed multi-view signatures in time-series modeling.

**Weaknesses:**

- [W1] **Questionable motivation of synthetic frequency classification experiments.** The first experimental section tests RFormer on two synthetic datasets with which the task is to classify input time-series based on their ground-truth frequencies. While L233-234 mentions the second setup in particular is designed towards testing the long-range reasoning capabilities of RFormer, but it is unclear whether this indeed the case. For the second synthetic dataset, in particular, how can identifying the frequency be a proxy for long-range reasoning when the frequency $\omega_0$ is used for $t < t_0$ only? The results on Figure 2 showing that methods that are "tailor-made for long-range time series modeling" (L254) such as Neural-CDE and Neural-RDE underperforming significantly also indicates that the designed task is not really representative of long-range reasoning.
\
\
More interesting questions to ask could be: What makes RFormer sample-efficient vs. vanilla Transformer particularly on the Sinusoidal dataset and not so much on the Long Sinusoidal dataset? What makes RFormer more robust to changes in sampling frequency compared to Neural-CDE and Neural-RDE? Table 1 of [A] shows Neural-CDE is also quite robust to dropped data, but is this characteristic not emergent for Sinusoidal and Long Sinusoidal datasets?

- [W2] **Missing analysis on interpolation methods.** By default, RFormer uses piecewise-linear paths for computing path signatures, but as mentioned in L140, it seems any continuous-time interpolation can be used. As such, it would be interesting to discuss (1) whether any other interpolation techniques can be deployed efficiently similarly to piecewise-linear paths and (2) if they lead to any boosts in predictive accuracy, but these discussions are missing in the current draft (i.e., is the currently used piecewise-linear interpolation "pareto-optimal" under performance-efficiency trade-offs?).

[A] Kidger et al., Neural Controlled Differential Equations for Irregular Time Series. NeurIPS 2020.

**Questions:**

- [Q1] **Large overlaps in representation space.** Basec on the multi-view signature formulation in Figure 1 and Equation 8, it seems the earlier observations would be covered by a large number of input tokens. As shown in the right plots in Figure 3, this would result in large "representational overlap" similarly to the oversmoothing phenomenon in graph representation learning [B]. Considering this, could restricting the global view to a few previous points rather than all previous points be a viable option? or does the theoretical and empirical robustness of RFormer to variable lengths and irregular sampling require that all previous points be covered in the global view?

- [Q2] **Discussion on input length of MLP and Transformer.** L108 states that the input length $L$ of the MLP and Transformer is fixed by assumption, but is this true? Sequences with different lengths can be processed in a single batch via padding.

[B] Rusch et al., A Survey on Oversmoothing in Graph Neural Networks. arXiv 2023.

**Limitations:**

The authors have adequately addressed the limitations in Appendix B.

---

> ### Author Rebuttal · Authors · 2024-08-06
>
> We thank the reviewer for the feedback, as it has been very relevant in updating our manuscript (especially the comment on oversmoothing). We hope that the reviewer will be satisfied with the additional experiments carried out and will be inclined to raise the score.
>
> **[W1.1] Motivation of synthetic frequency classification.**
>
> The idea of this experiment was to have the relevant information for the classification experiment at the beginning of the time series, as it would have to be propagated forward in order to correctly perform the classification. This is the type of task where we expect traditional recurrent models to fail due to vanishing gradients if the sequence is relatively long (2000 points in this case). This is likely the reason for the poor performance of Neural CDEs, which are a continuous-time analog of RNNs. However, we highlight that this was useful to test if RFormer was able to retain the long-range capabilities of the Transformer despite operating on a very compressed representation. This was further evaluated in a number of very long datasets (most notably EigenWorms), where RFormer was not only the fastest but also the most performant.
>
> If the reviewer would suggest an alternative that might be more suitable in their view, we would be more than happy to run additional experiments for this task.
>
> **[W1.2] Clarification on RFormer’s sample efficiency on Sinusoidal vs. Long Sinusoidal datasets.**
>
> Upon inspection, we found a bug in our code for this particular experiment, where we were not adding the time information to the dataset before passing it to RFormer (this was only the case in this experiment, due to an incorrect local version of the code). After re-running the experiment, we see similar sample efficiency gains from RFormer in this task for the same hyperparameters. Furthermore, we found that some additional tuning of the signature level in this new setting led to even greater sample efficiency gains (Thank you for this!). The results can be found in Figure 2 in the associated PDF. We have also included an experiment investigating the effect of signature hyperparameters in Figure 2 of the PDF as well.
>
> **[W1.3] RFormer's robustness to sampling frequency changes vs. Neural-CDE/RDE.**
>
> The theoretical reason behind why RFormer can process irregularly sampled sequences is shown in Proposition 3.1 (L175). In terms of the performance of Neural CDE and Neural RDE, we note that they experience a similar drop for this task as in the other tasks considered. To better understand how our model compares to NRDEs and NCDEs in settings in which these baselines perform well, we ran an additional random dropping on 15 experiments, which are shown in Table 4 of the PDF. We find that RFormer is not only better performing, but also several orders of magnitude faster than the rest of the SoTA benchmarks considered.
>
> **[W2] Missing analysis on interpolation methods.**
>
> We agree that investigating the impact of different interpolation methods in the multi-view signature computation is interesting. Our choice to use linear interpolation of data was due to the efficient signature computation of paths of this form. This efficiency is due to 1) linear interpolation being a *local* operation and 2) as noted in the Appendix (L644), the signature of piecewise-linear functions can be computed explicitly in terms of the data points. Continuous interpolations such as splines are non-local, meaning the computational burden is high for long time series. Deriving a convenient, computationally efficient method for computing the signature is also non-trivial.  As such, we could consider this approach to be "pareto-optimal", as the reviewer suggests.
>
> Since we focused our experiments on long time series, we decided against investigating this in our paper. However, based on your review, we will add a small section detailing this choice.
>
> **[Q1] Large overlaps in representation space.**
>
> We thank the reviewer for this excellent question. We have investigated this matter further, and this has led to some very interesting new insights that we feel should be brought to the attention of all the reviewers. For this reason, we have included our response to this question in our General Response.
>
> Additionally, we would also note that the multi-view signature transform is only an instance of a broader idea of presenting the model with information at different scales, which we found to improve performance (see Ablation in Appendix E.1). There are many other ways of providing this multi-scale information (such as restricting the global view to a few previous points rather than all previous points, as suggested by the reviewer), but this would be dataset and task-dependent. In the limit, we hope that these parameters can be tuned with the downstream task loss, but we leave this for future work.
>
> We have included a discussion of our hypothesis of why RFormer achieves substantial efficiency gains in our manuscript, as well as a related citation [1]. We thank the reviewer again for the very helpful comment on oversmoothing.
>
> **[Q2] Discussion on input length of MLP and Transformer.**
>
> Thank you for bringing this point to our attention. We agree with the reviewer that padding can be used as an ad-hoc solution to guarantee the same sequence length in the batch, even though it has been found that other models (recurrent and convolutional) sometimes struggle with this solution, see [2].
>
> We wanted to highlight this as a weakness of the Transformer model when comparing it to RFormer, which will always give the same representation to the model due to the flexibility of the signature. However, we will tone down our wording in accordance with the reviewer's suggestion.
>
> [1] Rusch, T. Konstantin, et al. "Graph-coupled oscillator networks." International Conference on Machine Learning. PMLR, 2022.
>
> [2] Dwarampudi, Mahidhar et al. "Effects of padding on LSTMs and CNNs." arXiv preprint arXiv:1903.07288 (2019).

---

> > ### Comment · Reviewer_wNkv · 2024-08-14
> >
> > Thank you authors for your time and commitment in preparing the rebuttal. All of my concerns have been addressed, and thus I increase my rating to 7.

---

> > > ### Author Response · Authors · 2024-08-14
> > > **Thank you for raising the overall and confidence scores**
> > >
> > > Thank you for your response and your very helpful comments, which have greatly improved our paper. We are glad that we could address your concerns and appreciate your decision to raise both the overall and confidence scores. Thank you again!

---

### Author Rebuttal · Authors · 2024-08-06

We would like to thank the reviewers for their valuable feedback. We have carefully reviewed their comments and incorporated their suggestions into a new version of the manuscript. Additionally, we are encouraged by the positive feedback provided by the reviewers on the novelty, efficiency, and performance of the proposed method, as evidenced by comments such as:

- Good novelty. [...] a novel approach [...] of great interest to the machine learning community.
- Great empirical performance. Experiments on a wide variety of real-world datasets show large improvements in both accuracy and efficiency, demonstrating strong utility of proposed multi-view signatures in time-series modeling.
- [...] the combination of rough path signatures with attention is a novel combination.
- Experiments on classification and regression tasks show that RFormer improves over existing models in terms of accuracy and is significantly more compute efficient.
- Overall the paper is well written and is easy to follow.
- The idea of using signature transform within an attention mechanism is interesting.

We believe RFormer constitutes a prime example of the general effort in combining effective modelling techniques from different research fields for addressing practical machine learning tasks, which has the potential to inspire further similar approaches.

In this general response, we aim to address some of the common questions among reviewers, as well as new insights gained during the rebuttal process as a result of excellent observations from the reviewers.

- **New Datasets and Baselines**

A common sentiment noted among reviewers was the need for additional datasets and baselines. To address this, we have implemented RFormer in 19 new datasets from the UCR time-series archive, consisting of 15 shorter time-series datasets and 4 long time-series datasets. We benchmarked against a number of new methods including SoTA methods for irregular temporal processing such as Neural {SDE, LSDE, LNSDE, GSDE} [1] and CRU [2], as well as SoTA state-space models (S5, LRU, and Mamba). For the irregular sampling experiments, we randomly dropped datapoints at rates of 30%, 50%, and 70%.

The results of these experiments can be found in Tables 4 and 5 in the accompanying PDF. RFormer consistently outperforms the Neural ODE-based methods on the 15 short time-series datasets and the state-space models on the long time-series datasets. We also include an additional comparison between CRU and RFormer in an irregularly sampled synthetic data setting (supported by CRU's hyperparameters validation in Table 1 and an analysis of training times in Table 3, both in the PDF). We present the results in Table 2 of the PDF. Note that shorter sinusoids with a smaller number of classes than the experiments in Section 4.1 are used here, which demonstrates that recurrent models perform well with short sequences in this synthetic example. Also, as well as demonstrating superior performance, our model is significantly faster than other continuous-time models, as shown in Appendix F.2 and Table 3 of the PDF.

- **Large Overlaps in Representation Space**

We thank Reviewer wNkv for raising the excellent connection between the representational overlap of the global signature transform and the oversmoothing phenomenon present in graph representation learning. We investigated this question deeply and attained new results which we feel give new insights into the performance of our model and should be brought to the attention of all reviewers.

In our experiments, we found that Transformers work better with more coarse representations of input data, and we believe that this is the reason behind some of the performance gains that we observe in RFormer. By coarsening the representation offered to the Transformer backbone, the model is able to learn better and faster.

However, as Reviewer wNkv points out, this only occurs if the intervals at which windows are taken are sufficiently large. Otherwise, the signatures may exhibit some form of "representational collapse". We note that in our experiments on very long time-series, we took the signature over windows that were very spaced apart, which prevented global representations from being too similar and seemed to yield better performance.

To offer a more quantitative evaluation of oversmoothing in this setting, we measured the Dirichlet energy by interpreting the temporal signal as a directed path graph. We compared different numbers of windows (from 2 to ~18k) of the global signature in the EigenWorms dataset. The results are shown in Figure 1 of the attached PDF. Interestingly, we found that the "elbow" of the Dirichlet energy corresponded to 30 windows in this dataset, which we found empirically to be the most performant setting. This hints at the idea of the Dirichlet energy being used for signature hyperparameter tuning as well.

We thank Reviewer wNkv again for this very interesting suggestion. We will add a dedicated section discussing these conclusions in the paper, as we believe it could shed light on the internal mechanisms of Transformers as well as motivate our method more strongly. We have also added some citations to works associated to oversmoothing.

- **Time-Series Forecasting**

To assess the model's ability to understand the dynamics of the time series, we have carried out an additional experiment of forecasting the next-step intraday volatility of an Apple Limit Order Book. The results are shown below:

|     | RFormer | Transformer | NRDE  |
|----------|---------|-------------|-------|
| **RMSE**     | 32.33   | 33.45       | 37.22 |



[1] Oh, Y., et al. Stable Neural Stochastic Differential Equations in Analyzing Irregular Time Series Data.Twelfth International Conference on Learning Representations.

[2] Schirmer, Mona, et al. "Modeling irregular time series with continuous recurrent units." International conference on machine learning. PMLR, 2022.

---

### Decision · Program_Chairs · 2024-09-25

**Decision:**

Accept (poster)

**Comment:**

This paper proposes a Transformer-based architecture for continuous-time signals, using multi-view attention on path signatures (from rough path theory) to capture both local and global dependencies. The authors effectively addressed the reviewers' concerns during the author-reviewer discussion phase. After the discussion, the majority of reviewers support the acceptance of this paper. AC agrees with the reviewers' positive opinions: this paper presents a novel approach by incorporating rough path theory into time-series representation learning, provides theoretical support for robustness, and empirically demonstrates the effectiveness of the proposed architecture. Although concerns from one negative reviewer remain, AC thinks the author rebuttal well addressed the concerns. For example, the authors added new comparisons with recent baselines as the reviewer suggested.

Therefore, AC recommends acceptance and highly encourages the authors to incorporate all experimental results (e.g., new datasets, new baseline models) and analyses from the author-reviewer discussion into the final manuscript, as they would further strengthen the contributions of this paper.